# Evolutionary paths toward multi-level convergence of lactic acid bacteria in fructose-rich environments
Naoki Konno [1] ✉, Shintaro Maeno[2], Yasuhiro Tanizawa[3], Masanori Arita [3], Akihito Endo [4] & Wataru Iwasaki [1,5,6,7,8,9] ✉

Convergence provides clues to unveil the non-random nature of evolution. Intermediate paths toward convergence inform us of the stochasticity and the constraint of evolutionary processes. Although previous studies have suggested that substantial constraints exist in microevolutionary paths, it remains unclear whether macroevolutionary convergence follows stochastic or constrained paths. Here, we performed comparative genomics for hundreds of lactic acid bacteria (LAB) species, including clades showing a convergent gene repertoire and sharing fructose-rich habitats. By adopting phylogenetic comparative methods we showed that the genomic convergence of distinct fructophilic LAB (FLAB) lineages was caused by parallel losses of more than a hundred orthologs and the gene losses followed significantly similar orders. Our results further suggested that the loss of *adhE*, a key gene for phenotypic convergence to FLAB, follows a specific evolutionary path of domain architecture decay and amino acid substitutions in multiple LAB lineages sharing fructose-rich habitats. These findings unveiled the constrained evolutionary paths toward the convergence of free-living bacterial clades at the genomic and molecular levels.

Convergent evolution has been a powerful clue to suggest non-random or predictable nature of evolution: both microevolution[1-6] and macroevolution[7-10]. Statistically significant similarity among independent evolutionary outcomes suggests the existence of shared selective pressures or evolutionary constraints among lineages[11,12]. To date, convergence has been studied at multiple levels including phenotypic, genomic, and molecular levels[13-17]. While many studies have focused on the evolutionary mechanisms leading to extant traits, studies on the intermediate evolutionary steps and processes have provided insights into the randomness and constraints on potential evolutionary paths from a past to a current state[18-23]. Those constraints on evolutionary paths have been observed by comparing distinct evolutionary paths toward convergence, i.e., the orders of multiple evolutionary events in different lineages. Evolutionary paths toward convergence can be different among lineages if the order of evolutionary events is random (the "evolutionary funnel" model), or similar if the intermediate evolutionary process is non-random or constrained

(the "evolutionary stream" model) (Fig. 1). "Evolutionary funnel model" suggests that there are potentially diverse evolutionary intermediates from the common ancestor to the converged evolutionary outcomes, while "evolutionary stream model" suggests that evolutionary constraints only allow specific evolutionary paths.

Constraints on intermediate paths of microevolution were studied by direct tracing of evolution or by measuring the fitness of artificial evolutionary intermediates[18-20,24-26]. Those microevolutionary studies showed that mutations on a gene accumulate following constrained paths, which reflect epistatic relationships among amino acid residues. On the other hand, macroevolutionary convergence in nature often takes so long that evolutionary intermediates are not generally observable. Phylogenetic comparative methods have been widely adopted to reconstruct long-term processes on a phylogenetic tree[27-29]. While a study on long-term phenotypic convergence supported stochastic paths, that is, the evolutionary funnel model[30], studies on genomic evolution have suggested an evolutionary

[1]Department of Biological Sciences, Graduate School of Science, The University of Tokyo, Bunkyo-ku, Tokyo, Japan. [2]Research Center for Advance Science and Innovation Organization for Research Initiatives, Yamaguchi University, Yamaguchi, Yamaguchi, Japan. [3]Department of Informatics, National Institute of Genetics, Mishima, Shizuoka, Japan. [4]Department of Nutritional Science and Food Safety, Faculty of Applied Bioscience, Tokyo University of Agriculture, Tokyo, Japan. [5]Department of Integrated Biosciences, Graduate School of Frontier Sciences, The University of Tokyo, Kashiwa, Chiba, Japan. [6]Department of Computational Biology and Medical Sciences, Graduate School of Frontier Sciences, The University of Tokyo, Kashiwa, Chiba, Japan. [7]Atmosphere and Ocean Research Institute, The University of Tokyo, Kashiwa, Chiba, Japan. [8]Institute for Quantitative Biosciences, The University of Tokyo, Bunkyo-ku, Tokyo, Japan. [9]Collaborative Research Institute for Innovative Microbiology, The University of Tokyo, Bunkyo-ku, Tokyo, Japan. ✉e-mail: konno-naoki555@g.ecc.u-tokyo.ac.jp; iwasaki@k.u-tokyo.ac.jp

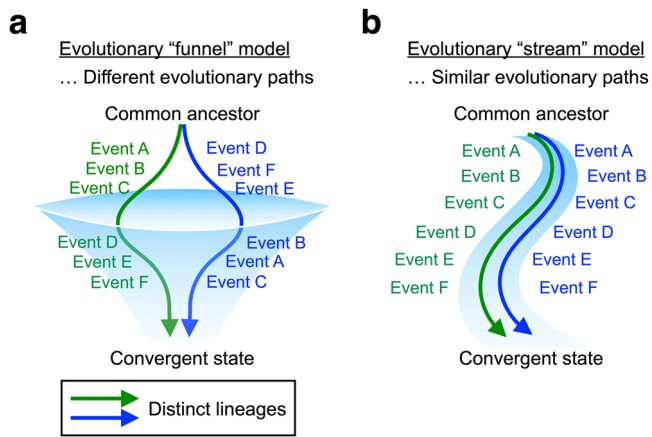

**a** Evolutionary "funnel" model
… Different evolutionary paths

**b** Evolutionary "stream" model
… Similar evolutionary paths

**Fig. 1 | Models for convergence through multiple evolutionary events. a, b** Given two distinct lineages show genomic or phenotypic convergence through multiple evolutionary events (A–F), the lineages potentially follow different evolutionary paths (the evolutionary "funnel" model) (**a**), i.e., experience events A–F in different orders or follow similar paths (the evolutionary "stream" model) (**b**).

stream model[23,31,32]. In addition, previous studies on convergent genomic evolution have been limited to the genome streamlining of organelles and endosymbiotic species such as mitochondria[23,31] and Buchnera[32]. Thus, it is still unclear whether a common order of evolutionary events is observed in the macroevolutionary convergence, especially of free-living species.

Fructophilic lactic acid bacteria (FLAB) are a group of Lactobacillaceae species sharing unique metabolic traits (explained below) and are reported to show clear convergent evolution in terms of phenotypic, ecological, and genomic traits[33]. FLAB consists of two phylogenetically distant genera, *Fructobacillus* and *Apilactobacillus* (excluding *A. ozensis*), which initially belonged to the families *Leuconostocaceae* and *Lactobacillaceae*, respectively[34,35]. Phenotypically, FLAB are a unique group of lactic acid bacteria (LAB) that grow poorly on glucose but well on fructose. Supplementation with external electron acceptors markedly improves the growth of FLAB on glucose[36]. Pyruvate, oxygen, and fructose are the major sources of the electron acceptors. In addition, plant-derived phenolic compounds are also possible electron acceptors[37]. Ecologically, FLAB are free-living clades found only in fructose-rich niches, including fruit surfaces, fermented fruits, flowers, and honeybee guts, which environments likely led to the convergent evolution[33,35,38]. Honeybee-based food products are a rich source of FLAB[39].

FLAB possess small genomes (<1.69 Mbp) and small numbers of coding DNA sequences characterized by the convergent absence of genes, especially for metabolism such as phosphotransferase system transporters[33]. The genomic characteristics of FLAB cannot be explained only by typical genome streamlining, as their gene repertoires are not similar to those of other related clades with small genomes[33]. A characteristic of the FLAB genome is the absence of *adhE* gene[40,41]. AdhE is an 850–900 amino-acid bifunctional enzyme that has alcohol dehydrogenase (ADH) and acetaldehyde dehydrogenase (ALDH) domains, and is one of the key enzymes for NAD/NADH cycling in the heterolactic phosphoketolase pathway[42]. FLAB use this pathway for glucose metabolism; thus, the lack of AdhE causes poor glucose growth if external electron acceptors for the oxidation of NADH in glucose metabolism are unavailable[33,43]. FLAB grow well on fructose, which functions both as a carbon source and an electron acceptor. In contrast, *adhE* is essential for all heterofermentative LAB, excluding FLAB[43]. Notably, the sole non-FLAB member in the genus *Apilactobacillus*, *A. ozensis*, grows on glucose and possesses both complete AdhE (877 amino acids) and partial AdhE (458 amino acids)[43]. In this way, FLAB have been defined as a group of lactobacilli sharing metabolic traits: (1) Growing well on fructose and requiring electron acceptors (e.g., fructose, oxygen, or pyruvate) to grow on glucose, (2) Metabolizing limited number of

carbohydrates, (3) Lacking many genes for glycometabolism, and (4) Lacking *adhE* fully or partially.

In this study, we first applied ancestral reconstruction of gene content to the recently accumulating genome resources of various Lactobacillaceae species and analyzed the intermediate evolutionary processes of genomic convergence toward FLAB. Then, we statatistically investigated which of the evolutionary funnel or stream model suits the gene-content evolution of FLAB. We further focused on the molecular evolutionary process of *adhE* gene loss in terms of domain loss and amino acid substitution patterns, and verified which evolutionary model is supported for the evolution at a molecular level. Our study shed light on the possibility of constraints on the macroevolutionary paths toward convergence of free-living species and the importance of systematic evolutionary analysis in revealing the driving forces and predicting the future of natural evolution.

## Results
### Reconstruction of gene-content evolution toward two FLAB clades

To analyze intermediate processes toward genomic convergence of the two FLAB lineages (*Apilactobacillus* and *Fructobacillus*), we reconstructed the gene content of ancestral species from that of extant species using Diversitree[44]. A function of Diversitree estimates the presence/absence of an ortholog group (OG) in each ancestral species, i.e., each internal node in a phylogeny, by maximizing the likelihood under a stochastic model (Mk model[45]) of gene gains/losses (Fig. 2a). This method requires the phylogeny and a profile of the presence/absence of an OG for every extant species, i.e., every tip of the phylogenetic tree. To prepare the required dataset, we retrieved representative genomes and a genome-based reference phylogenetic tree for all phyla of bacteria from Genome Taxonomy Database (GTDB r202), a database of a genome-phylogeny-based taxonomy of prokaryotes[46]. The reference phylogenetic tree was reconstructed from a concatenated multiple sequence alignment of 120 marker genes[46]. We then extracted reference genomes and a phylogenetic tree of all the 344 Lactobacillaceae species for which GTDB provides high-quality genomes (>95% completeness and <5% contamination). We employed stringent thresholds for genome completeness and contamination due to the risk of false negatives and false positives in detecting orthologous genes in our downstream analysis (explained below). We finally constructed a presence/absence profile for 2293 OGs using a gene annotation tool (KofamScan[47]) applied to the representative genomes of the 344 species. Because the estimation of ancestral gene content was uncertain, we repeated the inference of ancestral presence/absence 500 times for every OG with different model parameter sets pre-computed using a Bayesian method[44]. We eliminated the possibility of auto-correlation among the 500 model parameters to ensure that the repeatedly reconstructed ancestral states were independent of each other (Supplementary Fig. 1a).

We extracted OGs commonly and independently lost in the two lineages leading to FLAB clades, that is, OGs that were present in the common ancestor (LCA) of the two FLAB clades (LCAfa) but absent in the LCA of *Apilactobacillus* (LCAa) and that of *Fructobacillus* (LCAf) (Fig. 2b). We classified every reconstruction result for each OG into eight ($2^3$) scenarios corresponding to the presence/absence in LCAfa, LCAa, and LCAf and represented each scenario as a triplet of zero (absence) and one (presence). For example, a scenario in which an OG is present in LCAfa, absent in LCAa, and present in LCAf is represented by (1, 0, 1). We selected the most strongly supported scenario for each OG from the 500 reconstructions (Fig. 2c). We found that the supported scenarios for most OGs were robust against the independently reconstructed ancestral states, and found 137 OGs (55 metabolic and 82 non-metabolic OGs; metabolic genes were defined as set of OGs included in 'Metabolism' of KEGG Brite database[48]) commonly lost in the two lineages leading to the FLAB clades.

We further found that the overlap between the metabolic gene sets lost along the two evolutionary paths toward the FLAB clades (LCAfa-to-LCAa and LCAfa-to-LCAf) was larger than expected by chance, confirming the previously suggested metabolic convergence of FLAB[33] (Fig. 2d). The non-

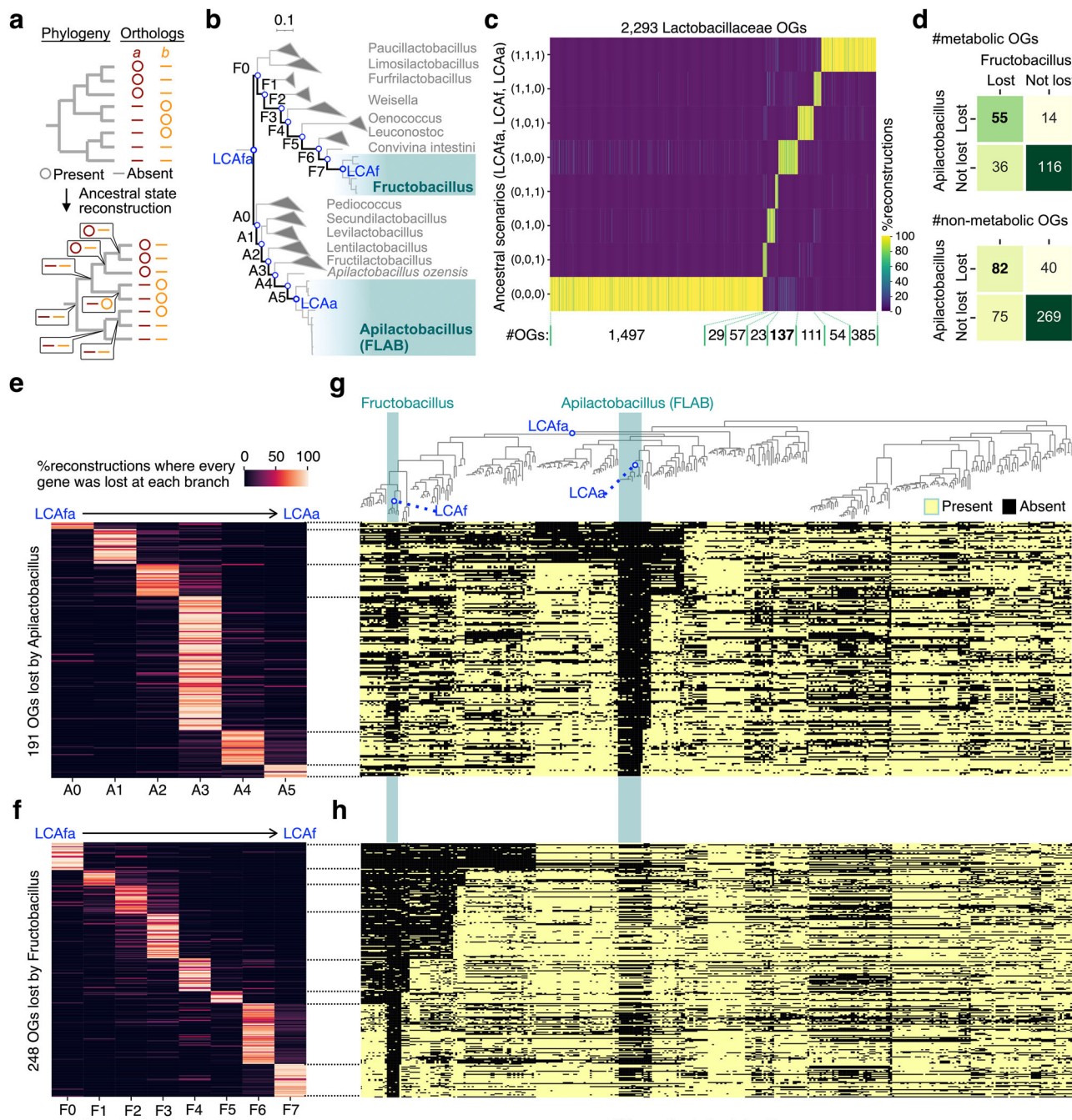

**Fig. 2 | The genomic convergence of FLAB involved shared losses of over 100 orthologs. a** A schematic diagram of ancestral reconstruction. Gene presence and absence at each extant or ancestral species are represented by circles and bars, respectively. In this example, the presence or absence of orthologs *a* and *b* for each internal node of the phylogenetic tree is estimated based on whether they are present or absent at each tip. Here, orthologs *a* and *b* are inferred to have been acquired once and twice (suggesting horizontal transfers for *b*), respectively. **b** The phylogenetic relationship of Lactobacillaceae species that are descendants of the latest common ancestor of the two FLAB clades. Paths from the common ancestor of two FLAB clades (LCAfa) to the common ancestor of each of them (LCAa and LCAf) were indicated with thick branches. F0-F7 and A0-A5 represent the branches of the tree from LCAfa to LCAf and LCAa, respectively. The scale bar indicates the branch length of 0.1. **c** Classification of Lactobacillaceae OGs based on inferred ancestral

scenarios. The heatmap shows the ratio of 500 ancestral scenarios inferred for each OG in every presence/absence pattern at LCAfa, LCAa, and LCAf. The number of OGs in each class corresponding to the major group of estimated scenarios was shown below the heatmap. **d** Contingency tables for metabolic and non-metabolic genes to test the significant overlap of OGs lost by *Fructobacillus* and *Apilactobacillus*. The color of each cell corresponds to the number in the cell. *P* values were $3.53 \times 10^{-15}$ and $4.39 \times 10^{-19}$ for metabolic and non-metabolic genes, respectively. Classifications of OGs lost in the lineages from LCAfa to LCAa (**e**) and LCAf (**f**) by the most likely branch where an OG was lost. The heatmap shows the ratio of ancestral scenarios supporting each branch as the branch where each OG was lost. Phylogenetic distributions of OGs lost in the lineages from LCAfa to LCAa (**g**) and LCAf (**h**). The OGs were sorted in the same order as **e** and **f**.

metabolic genes lost in the two lineages also showed significant overlaps, suggesting that the repertoires of non-metabolic genes converged under shared evolutionary pressures or constraints. A hierarchical clustering method previously used to detect the convergence of metabolic gene sets[33], clustered *Fructobacillus* and *Apilactobacillus* while the two genera are distantly related in the reference phylogeny, suggesting the convergence of metabolic gene repertoires (Supplementary Fig. 1b). Notably, the convergence of non-metabolic gene repertoires was not apparent in the dendrogram, because the two FLAB genera were not clustered (Supplementary Fig. 1c). The result suggests that we might have missed convergent gene-loss evolution without conducting ancestral reconstruction because of divergent gene gains and weak gene-content diversification in non-converging clades.

To reveal the order of gene loss in the two evolutionary paths toward FLAB, we next estimated the branch where the gene loss of each OG occurred (Fig. 2e, f). Among the branches in the paths toward *Apilactobacillus* and *Fructobacillus* (A0-A5 and F0-F7, respectively), we determined the gene-loss branch that was best supported by repeated ancestral reconstructions. We found that the supported branches were overall similar among the 500 reconstruction results and were consistent with the phylogenetic distribution of OGs in the extant species (Fig. 2g, h). Notably, the number of lost genes in *Apilactobacillus* evolution was especially large in branch A3, where the clade, including *Apilactobacillus* and *Fructilactobacillus* was diversified from *Lentilactobacillus*. Consistent with our results, *Fructilactobacillus* spp. are closely linked to specific niches, including flowers[49], beer[50], sourdoughs[51,52], and insects[34], and are known to have limited carbohydrate metabolic properties and small genomes[33,49,50,53]. These results suggest a phase of massive gene losses associated with the transition to their specific niches (e.g., insects and flowers) before the derivation of the clade, including *Fructilactobacillus* and *Apilactobacillus* (Supplementary Fig. 3). However, we did not observe such a peak in gene loss during *Fructobacillus* evolution.

In this way, we reconstructed the gene content of the evolutionary intermediates and the orders of gene gain/losses along with the paths toward the two FLAB clades to analyze whether the stream or funnel model fits the convergent evolutionary processes.

## Gene loss events toward FLAB clades supported the evolutionary stream model

To statistically assess whether the order of gene loss was shared between the paths leading to the FLAB clades, we evaluated the similarity between the gene loss orders of the two FLAB lineages. Here, the loss-order similarity was defined as the ratio of OG pairs lost in the same order between the two lineages in all pairs of commonly lost 137 OGs (Supplementary Data 3). We then calculated the null distribution of the loss order similarity by random shuffle of gene loss orders. Finally, we compared the observed similarity with the null distribution to test whether the observed score is significantly higher than expected by chance (Fig. 3a).

The results showed that the order of gene loss was significantly similar, supporting the evolutionary stream model for gene content evolution in FLAB (Fig. 3b). These trends were confirmed by visualizing the overlap ratio (Jaccard index) of the OGs lost in each branch of the two evolutionary paths (Supplementary Fig. 2a). To interpret the evolutionary paths shared between the lineages leading to the two FLAB clades, we mapped 137 commonly lost OGs onto functional categories defined by the COG database[54]. By sorting the functional categories according to the average timing of gene loss, we found that most functional categories (14 of 18) were lost in a similar order (Fig. 3c, d). Functions lost early in both lineages included secondary metabolism and defense mechanisms, which tend to be encoded in accessory genomes[55-57]. Function unknown genes ("Function unknown" and "General function prediction only") also tended to be lost early. Because genes in accessory genomes and genes of unknown function are often non-essential, these results suggest that non-essential genes were lost early in both convergent evolutionary paths.

In contrast, four of the 18 functional categories did not show similar gene loss orders. These four categories contained 44 OGs, with 37 related to

*carbohydrate transport and metabolism* or *amino acid transport metabolism*. Notably, the two functions' order of loss was reversed between the two lineages. These trends were confirmed by mapping OGs onto pathways defined in the KEGG database[48] (Supplementary Fig. 2b). This result indicates that the gene-loss evolution of these two metabolic pathways followed the stochastic evolutionary funnel model, where different order of gene losses would have led to the same destination, that is, the loss of both functions.

The phylogeny of 344 Lactobacillaceae species was extracted from the maximum-likelihood genome phylogeny provided in GTDB[58] and further confirmed by re-inferring a phylogeny from the GTDB marker gene sequences of Lactobacillaceae species (Supplementary Fig. 4). We noted that the phylogenetic location of the *Pediococcus* clade was different from that reported in a previous study, likely because of the selection of the marker gene sets[59] (Supplementary Fig. 5a). While the Ultrafast bootstrap values of the relevant branches in our tree were 96%, the tree topology may have umbiguities. When we changed the position of the *Pediococcus* clade (Supplementary Fig. 5b), the significantly large overlap of the commonly lost gene sets was robustly observed ($P$ value = $6.8 \times 10^{-32}$) and the tendency of the gene-loss order similarity was also kept observed but without statistical significance after removing or regrafting *Pediococcus* clade (Supplementary Fig. 5c–f).

In summary, convergent gene losses toward FLAB were mainly consistent with the evolutionary stream model, except for genes involved in the metabolism of carbohydrates and amino acids.

## Non-FLAB lactobacillaceae clades in fructose-rich environment convergently lost *adhE*

Next, we focused on the loss of the *adhE* gene as a key convergent gene loss event toward the emergence of the two FLAB clades. To comprehensively detect and compare *adhE* loss processes that occurred during convergent evolution, we first examined whether other Lactobacillaceae clades also lost *adhE* under selective pressures similar to those of the two FLAB clades. Given a previous study's report that certain Lactobacillaceae species harbor partial *adhE* genes[33], we employed a sequence similarity-based search to sensibly detect these partial genes. This was achieved by querying complete *adhE* genes, detected from all bacterial clades through profile hidden Markov model (pHMM) searches (Supplementary Fig. 6a). We searched *adhE*-like genes by HMM profile searches against all high-quality genomes retrieved from GTDB with low contamination and high completeness (Supplementary Fig. 6b, c). After excluding markedly short or long sequences, we detected 4833 *adhE*-like genes from diverse bacterial phyla (Supplementary Fig. 6d). We then used this comprehensive *adhE*-like gene dataset as a query for sequence similarity searches by MMSeqs[60]. Here we set the threshold of sequence identity and coverage both as 45%. The thresholds were chosen for the sensitive detection of partial *adhE* genes. As a result, we identified 399 *adhE*-like genes in 369 non-FLAB Lactobacillaceae species (Supplementary Fig. 6e, Supplementary Data 1). Among the genes, only 287 genes (71.9%) were as long as typical *adhE* genes (800–1000 aa), and 77 genes (19.3%) were substantially shorter (350–600 aa) (discussed later).

Our extensive searches unveiled that multiple species in at least eight non-FLAB genera of Lactobacillaceae lost the *adhE* gene or had substantially shorter *adhE*-like genes. This suggests that *adhE* was repeatedly lost in non-FLAB clades neither *Apilactobacillus* nor *Fructobacillus* (Fig. 4a; Supplementary Data 2). Most non-FLAB clades lacking *adhE* were homofermentative species in which *adhE* was not essential. Notably, the homofermentative species missing *adhE* tend to miss pyruvate formate lyases, which are known to convert from pyruvate to acetyl-CoA[61] (Supplementary Fig. 6f). On the other hand, pyruvate formate lyase is not conserved in heterofermentative species as reported previously[59]. These findings suggests that the acetyl-CoA-to-ethanol conversion by AdhE in homofermentative species can be associated with activity of pyruvate formate lyase. Moreover, the occurrence of *adhE* loss in non-FLAB was suggested to be non-random because significantly high proportions of those non-FLAB species (30 of 87) were isolated from bee-associated

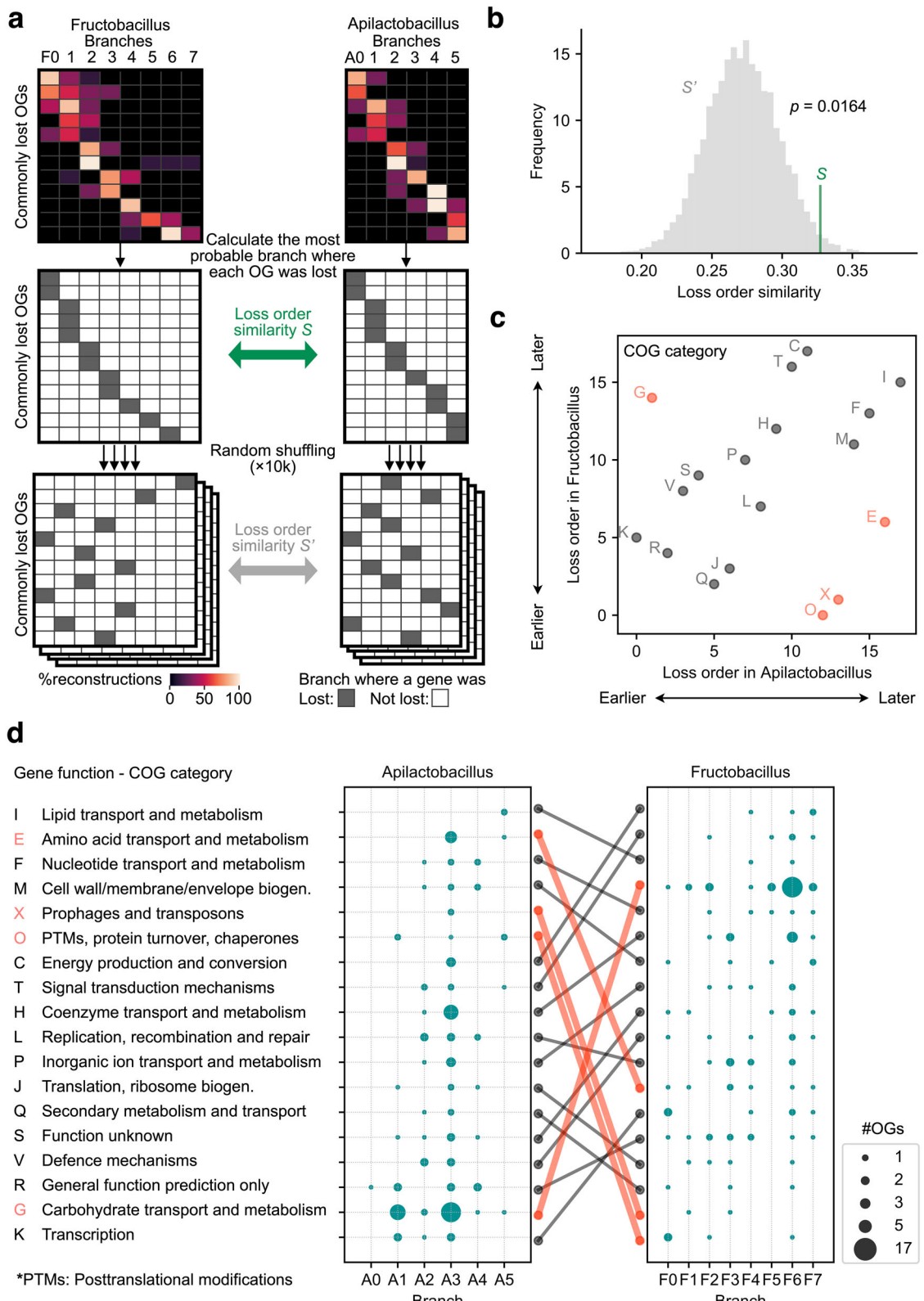

**Fig. 3 | Significantly similar yet partially divergent gene loss orders in the evolutionary paths of two FLAB clades. a** The schematic diagram of a statistical test method to detect the significance of gene loss order similarity between the two FLAB lineages. **b** The result of the statistical test depicted in (**a**). The green vertical line segment represents the observed score of gene loss order similarity calculated for 137 OGs commonly and independently lost by FLAB, while the grey histogram illustrates the null distribution of the score. The loss order similarity was defined as the ratio of OG pairs lost in the same order between LCAfa-to-LCAa and LCAfa-to-LCAf lineages. **c** Comparison of the relative loss order of COG functional categories in the two FLAB lineages. Each dot represents the order of average gene loss timings of each functional category in each lineage. Red and grey dots indicate functions that showed markedly different or similar relative timing of gene losses, respectively. **d** COG-functional-category-wise counts of OGs lost at each branch in paths toward the two FLAB lineages. The dot size indicates the number of lost OGs. Only the OGs commonly lost in the two lineages were included in the counts. The COG functional categories were sorted by the average timing of gene losses in each lineage. Red and grey edges connect the same functions, corresponding to functions represented as red and grey dots in (**c**).

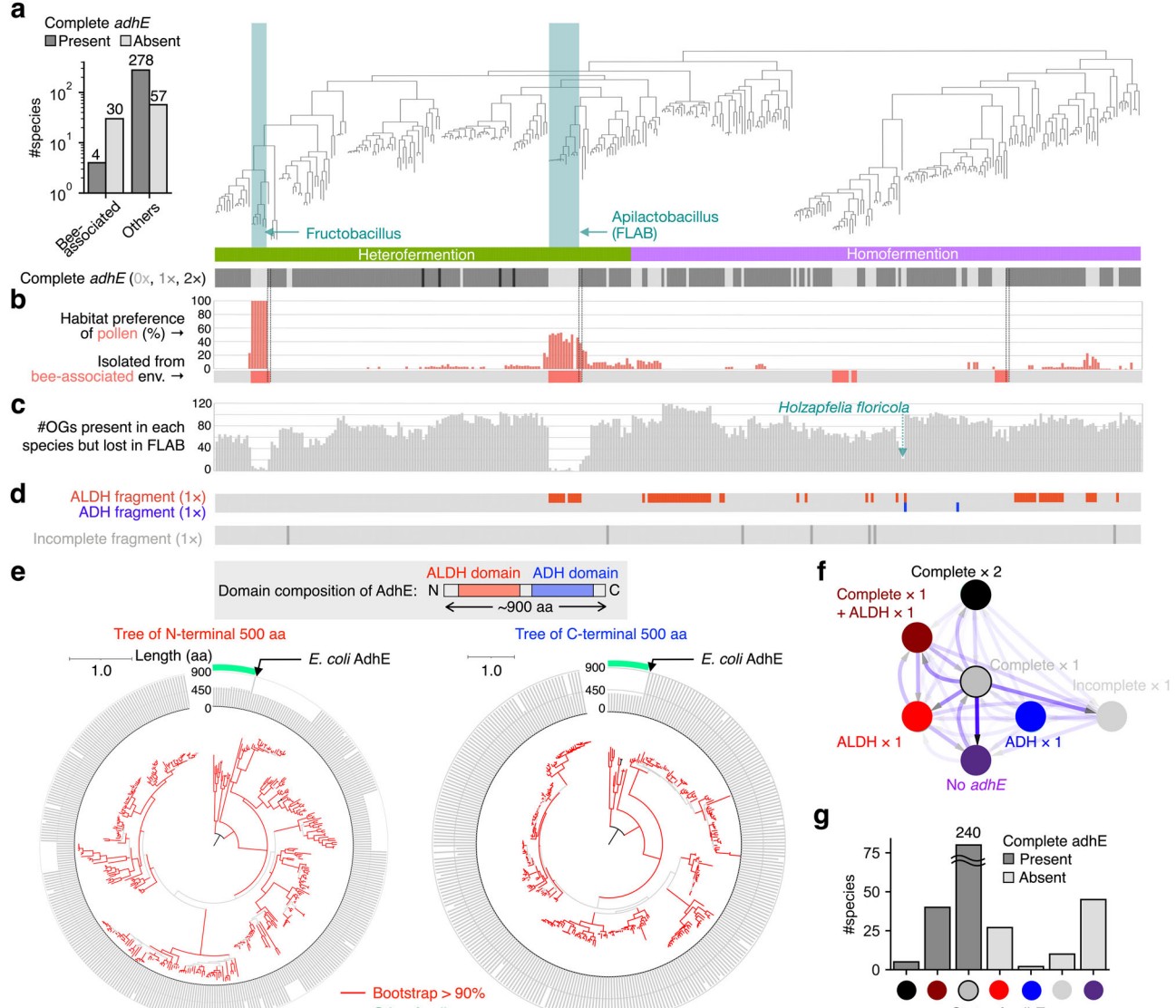

**Fig. 4 | Repeated loss of *adhE* in fructose-associated species and the tendency in the domain loss order. a** *adhE* is repeatedly lost in bee-associated clades. The left bar graph indicates the number of species with and without a complete *adhE* sequence by isolation site (bee-associated or others). The right panel shows the phylogenetic distributions of hetero/homofermentative species and complete *adhE* in a genomic phylogeny of 344 Lactobacillaceae species with a high-quality representative genome. The copy number of complete *adhE* is indicated as the color thickness of the heatmap. **b** Phylogenetic distribution of species in fructose-rich environments. Habitat preference scores for pollen environment and species isolated from bee-associated environments were represented by a bar graph and a color strip, respectively. **c** The phylogenetic distribution of the number OGs which are present in each species' genomes and commonly lost in the evolutionary paths toward two FLAB clades (LCAfa-to- LCAf and LCAfa-to- LCAa in Fig. 2b). **d** The phylogenetic distribution of partial *adhE*-derived genes which were classified into ALDH fragment, ADH fragment, and middle fragment. **e**, Gene phylogenies of N-terminal and C-terminal halves of complete and partial *adhE* in Lactobacillaceae and the amino acid lengths of their whole sequences. **f** State transition rates among the repertoire of complete and partial *adhE* across the whole Lactobacillaceae phylogeny. **g** Number of extant Lactobacillaceae species with each combination of complete and partial *adhE*. The X-axis labels represented as colored circles correspond to the labels in (**f**).

environments, which are major isolation sources of *Fructobacillus* and *Apilactobacillus* (Fig. 4a; Chi-square test, $p = 1.14 \times 10^{-20}$, $2.30 \times 10^{-20}$, and $1.83 \times 10^{-20}$ for all, heterofermentation, and homofermentation species, respectively). Three of the four bee-associated non-FLAB clades included at least one basal clade possessing *adhE*, which suggests that the *adhE* genes were lost after inhabiting bee-associated environments (Fig. 4b). Not only the isolation sites, we also analyzed habitats of each Lactobacillaceae species based on published datasets of shot-gun metagenome sequencing. Using a previously developed pipeline[62], we showed the enrichment of non-FLAB Lactobacillaceae species missing complete *adhE* in pollen (Fig. 4b; Phylogenetic ANOVA, $p = 0.002$), which is also known as a fructose-rich environment and a habitat for FLAB.

We also found that the genomes of non-FLAB species without *adhE* convergently lost a part of the 137 OGs that were commonly lost from the two FLAB clades (Phylogenetic ANOVA, $p = 0.006$ and 0.0002 for homo- and hetero-fermentation species, respectively) (Fig. 4c). This shared gene loss also supports the notion that non-FLAB clades without *adhE* were subjected to similar selective pressure with FLAB clades. A clear example of such a non-FLAB species is the homofermentative *Holzapfelia floricola*, isolated from flowers[63] and has a gene content similar to that of FLAB species[43].

Collectively, our results showed that multiple non-FLAB Lactobacillaceae clades, whose habitats are similar to those of FLAB, convergently lost *adhE* and other genes, likely under shared pressures.

### Repeated domain-wise loss of *adhE* also followed the evolutionary stream model

As previously mentioned, AdhE is a bifunctional enzyme with ADH and ALDH domains, and at least ten species of *Apilactobacillus* only have a partial *adhE* gene with the ALDH domain (Supplementary Data 2). This implies that the *adhE* gene was lost in a domain-wise manner in *Apilactobacillus*, where the ADH domain was lost first. Although we could not find partial *adhE* genes in *Fructobacillus* (Supplementary Fig. 6a; see section "Methods"), we found that many non-FLAB Lactobacillaceae species contained a partial (350–600 aa) *adhE* gene as described above (Supplementary Fig. 6e). Therefore, we examined whether domain-wise losses occurred in the non-FLAB Lactobacillaceae clades that lost *adhE* and whether they followed the same domain-loss order (i.e., the evolutionary stream model).

First, we carefully discriminated (partial) *adhE* genes from their paralogs because our sensitive searches of *adhE*-like genes could identify non-*adhE* genes only with ALDH or ADH domain. We constructed phylogenetic trees of the N- and C-terminal halves of all 399 *adhE*-like genes in the non-FLAB Lactobacillaceae species. Using the *adhE* gene of *E. coli* as an outgroup, we selected 372 genes, including 85 partial genes distributed in multiple clades, as complete and partial *adhE* (Fig. 4d, e and Supplementary Fig. 7). A sequence similarity network of 399 *adhE*-like genes also indicated that the 372 genes belonged to the same gene family cluster as *adhE* (Supplementary Fig. 6g). In both gene phylogenies and similarity networks, 85 partial genes were separated into different clades or clusters, indicating that domain-wise losses repeatedly occurred in Lactobacillaceae (Fig. 4e, Supplementary Fig. 6h).

We then classified the 85 partial *adhE* genes into three classes of "ALDH-only," "ADH-only," and "incomplete fragment" by analysis of the multiple sequence alignment (an incomplete fragment typically contains one domain and a part of the other domain). Interestingly, as many as 70 of the 85 partial *adhE* genes were ALDH-only fragments, while only five and ten were ADH-only and incomplete fragments, respectively. Thus, in both *Apilactobacillus* and non-FLAB Lactobacillaceae clades, *adhE* was lost in a domain-wise manner, and an ALDH-only fragment was the major evolutionary intermediate (Fig. 4d and Supplementary Fig. 7). For example, *Apilactobacillus apinorum* and *Lacticaseibacillus thailandensis* are completely missing *adhE*, while their sister groups possess one ALDH fragment only, suggesting ALDH fragment was the intermediate state of *adhE* loss in their lineages. We also found that ALDH-only fragments were conserved across species even after the loss of the ADH domain, suggesting that the ALDH-only fragments alone contributed to fitness (Fig. 4d). In contrast, the ADH-only fragments were not conserved across species. Three species (*Companilactobacillus zhachilii, Lacticaseibacillus saniviri, Lentilactobacillus kefiri*) were annotated to have both ALDH-only and ADH-only fragments in tandem, possibly caused by sequencing errors, because a complete *adhE* gene was found in all other strains of the three species (Supplementary Fig. 8). Therefore, we assumed that these three species possessed a complete *adhE* gene throughout the following analysis.

To quantitatively compare the frequencies of the *adhE* domain-wise losses in different orders, we estimated the transition rate parameters among the five states (complete, ALDH-only, ADH-only, incomplete, and no *adhE*) and two additional states because some species had another *adhE* gene in addition to an ALDH-only domain or a complete *adhE* gene (Fig. 4f). In the state transition model, we allowed transitions in which both ALDH and ADH domains are lost simultaneously because those transitions are possible by single mutations (e.g., nonsense mutations at upstream regions). We found that the transition rate from complete *adhE* to an ALDH-only fragment was 6.7 times faster than that to an ADH-only fragment. The results also revealed a substantially high rate of gaining an ALDH-only fragment by species with complete *adhE*, whereas none of the species with complete *adhE* gained the ADH fragment (Fig. 4f, g). We also found that the rate of a transition from "no AdhE" to "one complete AdhE" was non-zero, suggesting *adhE* genes could be horizontally transferred.

In summary, *adhE* loss occurred repeatedly in a domain-wise manner in Lactobacillaceae, initiating with the ADH domain loss. The finding suggests the decay of *adhE* domain architecture follows the evolutionary stream model.

### Amino-acid substitutions suggest the benefit of a specific intermediate step of *adhE* evolution

To reveal the molecular basis of how an ALDH-only fragment of the *adhE* gene can be beneficial and lead to constrained evolutionary paths, we investigated the evolution of amino acid sequences in ALDH domains. AdhE is known to function by forming a filament-like self-assembled complex called spirosome, where ALDH and ADH domains of different AdhE molecules interact[64], and substrate channeling of an intermediate product, acetaldehyde, between them enhances their enzymatic activities[65,66].

We constructed a molecular phylogenetic tree of the ALDH domain sequences encompassing the complete *adhE* genes and ALDH-only fragments. Then, we reconstructed the ancestral amino acid sequences for every internal node of the tree and extracted mutations specific to the ALDH-only fragment clades (Fig. 5a). Notably, some mutations were repeatedly observed, specifically in the ALDH-only fragment clades (Supplementary Fig. 9a). We found 15 sites that were the most variable among the ALDH-only fragments, but were conserved in the complete *adhE* genes (Fig. 5b, c). In particular, sites 105 and 451 accumulated specific substitutions in ALDH-only fragments (Fig. 5d). We confirmed that the estimation of ancestral sequences was supported by overall high posterior probabilities (Supplementary Fig. 9b).

To infer the characteristics of the sites with fragment-specific mutations, we mapped the 15 site variables in the ALDH-only fragments onto a previously reported AdhE complex structure (PDB:6ahc[65]) and found that these sites were enriched at the interface between the ALDH and ADH domains in the spirosome (Fig. 5e). In particular, sites 105 and 451 formed ionic bonds with another AdhE molecule (Fig. 5f). We statistically confirmed that the ratios of mutations in the ALDH-only fragments were significantly larger at the ALDH-ADH interaction interfaces than at other sites (Fig. 5g). These results strongly suggest that ALDH domains translated from ALDH-only fragments do not form spirosomes and do not require substrate channeling.

In contrast, we found that the $NAD^+$-binding sites in the ALDH-only fragments were strongly conserved, similar to those of the complete *adhE* genes (Fig. 5h and Supplementary Fig. 9c). These results suggest that the ALDH-only fragments retained the enzymatic activity of ALDH even after the ADH domains were lost. Notably, some sites in the ALDH-only fragments showed markedly better conservation than complete *adhE*, suggesting ALDH-only fragments convergently acquired point mutations (Supplementary Fig. 9d). This raised a possibility that ALDH-only fragments gained specific functions after domain loss, as reported for other protein families[67,68].

## Discussion

In this study, we dug into multiple LAB clades that convergently adapted to a fructose-rich environment to investigate the similarity of their long-term evolutionary paths at multiple levels. At the gene-content evolution level, we raised a possibility that the loss of as many as 137 genes occurred in significantly similar orders, although the result depends on the reference phylogeny topology. At the domain architecture evolution level, the repeated losses of *adhE* followed the same order of domain-wise losses as the first loss of the ADH domain. At the amino acid sequence level, we also observed a convergent accumulation of point mutations at the substrate-channeling interfaces of AdhE. The results suggesting evolutionary stream models at multiple scales illuminate the non-random nature and predictability of intermediate processes of evolution.

The similarity in the gene loss order based on the GTDB phylogeny was remarkable. While the evolution of endosymbiotic organisms and organelles follows similar evolutionary paths, FLAB are free-living and have experienced habitat changes[23,31,32]. As a recent study showed the general predictability of bacterial species that will gain/lose a gene in the future based

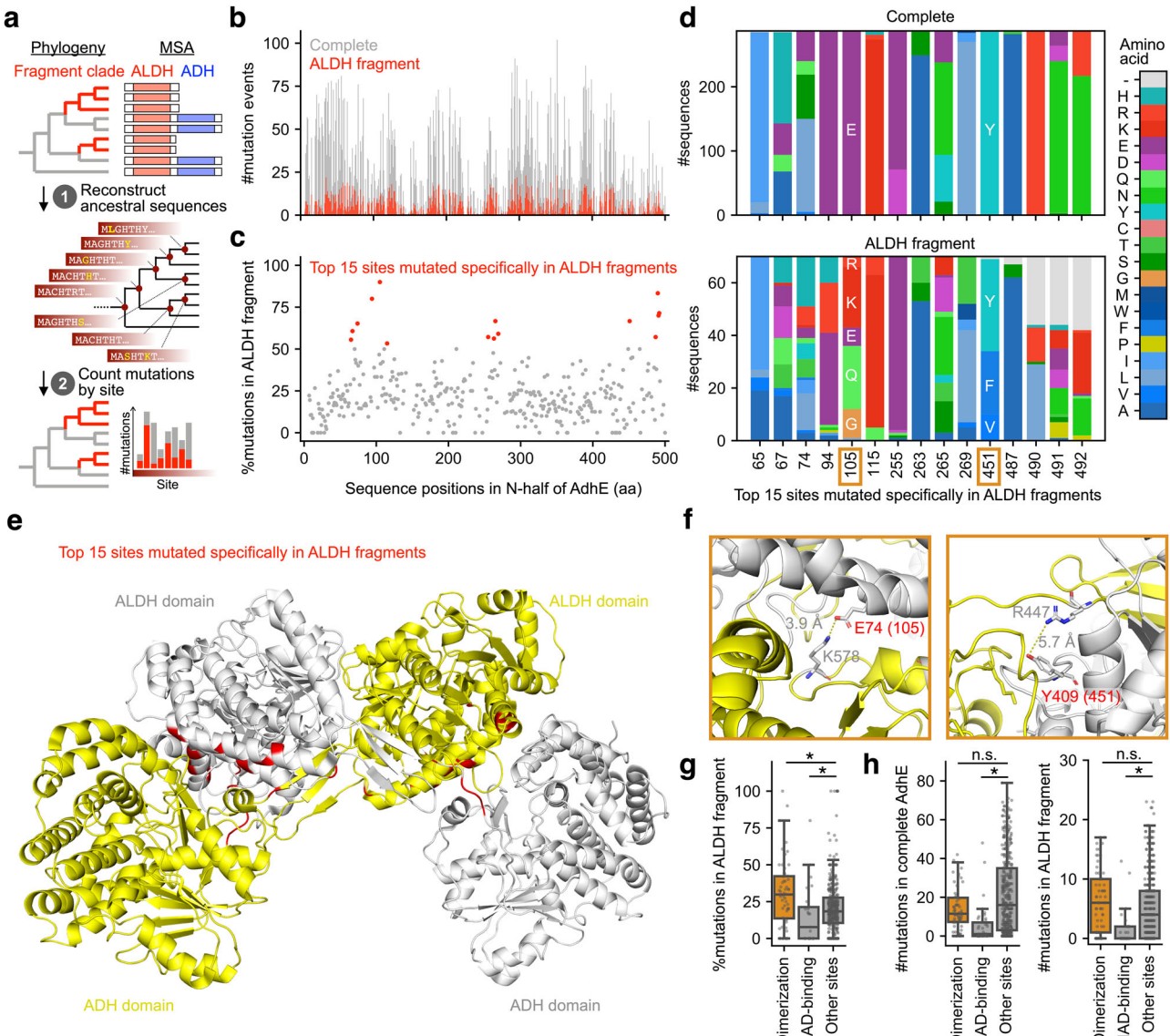

**Fig. 5 | Mutation history suggests fragmental AdhEs have lost channeling ability but retain enzymatic activity. a** A schematic diagram to detect mutations specific to partial *adhE* missing ADH domains. **b** Estimated number of amino acid substitutions that occurred on the gene tree of *adhE* at every sequence position in the N-terminal half of AdhE. The grey and red parts of every bar indicate the number of substitutions of complete AdhEs and ALDH fragments, respectively. **c** Enrichment of substitutions in ALDH fragments at every sequence position in the N-terminal half of AdhE. Each dot indicates the ratio of substitutions that occurred in ALDH fragments at a site of AdhE. Only sites mutated ≥5 times were indicated here. Red dots indicate the top 15 sites that tended to be substituted specifically in ALDH fragments. **d** Amino acid composition of the 15 sites in complete AdhEs and ALDH fragments. Amino acid characters are indicated for sites 105 and 451, which were focused on in (**f**). **e** A previously reported complex structure of AdhE dimer (PDB:

3ahc)[37]. The two AdhE molecules were colored white and yellow. The location of the 15 sites was colored red in each molecule. **f** Zoom-in diagrams of the AdhE dimer structures focusing on the two of 15 sites in the AdhE molecule colored white. **g** Enrichment of substitutions in ALDH fragments for AdhE-AdhE interaction sites, NAD-binding sites, and others. Each dot indicates the ratio of mutations that occurred in ALDH fragments at each sequence position. Box plots indicate the distributions of the data points. Asterisks indicate statistical significance by Mann–Whitney *U*-test ($p < 0.05$). **h** Number of substitutions that occurred in complete AdhEs and ALDH fragments for AdhE-AdhE interaction sites, NAD-binding sites, and others. Each dot indicates the number of substitutions that occurred in ALDH fragments at each sequence position. Box plots indicate the distributions of the data points. Asterisks and "n.s." indicate the significant and not significant differences by Mann–Whitney *U*-test ($p < 0.05$), respectively.

on current gene content information[69], there can be non-random or predictable patterns in the gene gain/loss order of diverse microbes. In the present study, both FLAB lineages independently lost non-essential genes before inhabiting fructose-rich environments, where specific genes were previously suggested to be commonly lost (e.g., phosphotransferase system[33]). Thus, the mechanism of the suggested evolutionary stream can be the gene essentiality affecting firstly lost genes and the following similar selective pressures in specific niches (e.g., insects and flowers) causing shared gene losses. Notably, gene loss patterns in *carbohydrate transport and metabolism* and *amino acid transport metabolism* did not follow the same

pattern in the two FLAB clades (Fig. 3d). In other words, the convergent gene losses of these two functional categories followed an evolutionary funnel model. While the ancestors of *Apilactobacillus* markedly lost carbohydrate metabolism genes during the homo-to-heterofermentation transition (first branch: A1), the ancestors of *Fructobacillus* lost these genes mainly in the later stages (F6). *Fructobacillus* mainly lost genes for amino acid metabolism at or soon after the homo-to-heterofermentation transition (F0–3). Although glycolysis would have become unnecessary after the transition to heterofermentation, our results suggest that genes for carbohydrate metabolism can be stochastically retained after the homo-to-

heterofermentation transition. Their retention may be because these two pathways are still advantageous in changing environments or at least not disadvantageous. It should be noted that the phylogenetic tree topology of Lactobacillaceae needs to be examined with the growing genome datasets in the future.

In addition to gene content evolution, we identified common evolutionary paths of the AdhE domain architecture in fructose-rich environments, where the ADH domain was first lost. This clear evolutionary trend is especially notable given that domain architecture convergence was reported to be rare in general[70]. It should be noted that the domain-loss frequency of proteins is generally high at both the N- and C-termini[71], and the bias of the domain-loss order is not likely attributed to their relative positions.

Our amino acid sequence analysis suggests that the loss of *adhE* genes tends to proceed through an ALDH-only fragment as an evolutionary intermediate. Our hypothesis to explain the order of domain loss is that ALDH fragments are less toxic than ADH fragments, and ALDH fragments can be beneficial for detoxifying aldehydes. Substrate channeling of AdhE has been reported to be critical for the forward reaction (i.e., conversion from acetyl-CoA to acetaldehyde) catalyzed by the ALDH domain[65]. According to the previous study, the acetaldehyde-producing activity of the ALDH domain was decreased by disrupting spirosome formation, while ethanol-to-acetaldehyde conversion activity of the ADH domain was not affected. Thus, we reasoned that ALDH fragments would have less activity to produce toxic acetaldehyde than ADH fragments.

We further hypothesized that the conserved ALDH-only fragments may contribute to fitness by removing the toxic aldehydes generated from other pathways (e.g., acetaldehyde generated from pyruvate via acetyl-CoA[72] and formate generated by pyruvate-formate lyases[61]), because active sites were still significantly conserved in ALDH-only fragments (Fig. 4h). Indeed, the ALDH-only fragment of *Apilactobacillus kunkeei* has been reported to catalyze a reverse reaction (the enzymatic activity for the forward reaction has yet to be investigated)[43]. Notably, the detoxification activity can be acquired before losing ADH domains. Moreover, we found species with both ALDH-only fragments and complete *adhE*, implying that ALDH fragments might contribute to fitness even in the presence of complete *adhE*. A previous study has suggested that the expression of complete AdhE can be toxic to bacteria[73]. Therefore, ALDH fragments may catalyze a reverse reaction to eliminate the acetaldehyde leaked from the AdhE spirosome and/or produced by other pathways as an intermediate product. Consistently, a eukaryotic clade, *Polytomella*, has two *adhE* genes, one of which has an ADH domain with very low activity, and an ALDH domain that preferentially catalyzes the acetaldehyde-to-acetyl CoA reaction[74]. Here, we focused on the evolution of *adhE* because loss of the gene characterizes FLAB. Nevertheless, we can generally apply the analysis methodologies to other multidomain proteins, which may further reveal the evolutionary patterns of domain architecture decay.

Collectively, our findings on stream-model convergent evolution at multiple levels extend our knowledge of the constrained evolutionary paths in the long-term evolution of free-living microbes. So far, convergent evolution at different levels has been studied using separate model clades[18–20,23–26,31,32]. Thus, analyzing the evolution of lactic acid bacteria will also lead us to find the interplay between evolutionary events at different levels of convergence, such as domain architecture evolution contingent on the losses of other specific genes. This study also established lactic acid bacteria in a fructose-rich environment as a model system to scrutinize convergent evolution at multiple levels and their interplays.

## Methods
### Dataset
We retrieved all the protein sequences of every representative genome and a reference phylogeny from GTDB r202 on April 28, 2021. The datasets covered 45,555 bacterial species of all the phyla defined in GTDB. A tree of Lactobacillaceae species (369 species) was extracted from the whole reference phylogeny using the *TreeNode.prune* function in ete3 toolkit 3.1.2

(ref. 75), preserving the branch lengths (with an option "preserve_branch_length = True"). We also extracted a tree of Lactobacillaceae for 344 species with high-quality representative genomes (defined as >95% completeness and <5% contamination throughout this study). We used representative genomes and phylogenies only of Lactobacillaceae throughout this study, except for searching complete *adhE* genes from all bacterial phyla (explained later). 16S rRNA gene sequences of 336 of the 344 species were downloaded from GTDB r202.

### Reconstruction of ancestral gene content
To reveal the order of gene loss in *Fructobacillus* and *Apilactobacillus* lineages, we reconstructed gene gain/loss scenarios (i.e., gene presence/absence at each internal node in the reference phylogeny) for every ortholog group. We firstly conducted Hidden Markov Model-based ortholog annotation for all the 344 high-quality representative genomes of Lactobacillaceae by KofamScan v1.3.0 (ref. 47). Using the results of the ortholog annotation for the Lactobacillaceae genomes by KofamScan, we first prepared an ortholog table representing the presence/absence (represented as one and zero, respectively) of each ortholog group for each of the 344 Lactobacillaceae species with high-quality representative genomes. Then, we estimated gene gain/loss rate parameters for every ortholog group under the two-state Mk model[45] using Markov chain Monte Carlo (MCMC) sampling implemented in Diversitree v0.9.16 (ref. 44). Here, we used an ultrametric Lactobacillaceae tree converted from the GTDB reference phylogeny by chronos function in ape 5.6.2 (ref. 76) package. For MCMC sampling, we initialized model parameters by maximum likelihood estimation and set the burn-in and sampling interval of MCMC sampling to 500 and 10, respectively (we confirmed negligible autocorrelation of estimated parameters under this setting (Supplementary Fig. 4)). This yielded 500 MCMC samples with gene gain/loss rate parameters for each ortholog group. Finally, we estimated the gene gain/loss scenarios for each of the 500 MCMC samples using the maximum-likelihood joint reconstruction in Diversitree v0.9.16.

### Test of gene loss order similarity
To evaluate the similarity between the evolutionary processes of the two FLAB clades (*Fructobacillus* and *Apilactobacillus*), we tested whether the order of gene loss was significantly similar. We first listed genes lost during evolution from the common ancestor of the two FLAB clades (LCAfa) to that of *Apilactobacillus* (LCAa) or that of *Fructobacillus* (LCAf) based on the best-supported gene gain/loss scenario for every ortholog. We then estimated the gene-loss branches for each ortholog, branches where each ortholog was lost, by selecting the most frequently supported scenario. To determine the most frequently supported scenario for every ortholog, we only considered scenarios in which gene loss occurred once during the evolution from LCAfa to LCAa/LCAf (Fig. 4a). Next, we calculated the similarity index $S$, defined as the ratio of ortholog pairs lost in the same order between the two lineages to all pairs of orthologs lost in both lineages. To obtain the null distribution of $S$, we calculated the same index ($S'$) after randomly shuffling the correspondence between the orthologs and loss branches. We repeatedly shuffled and calculated $S'$ 10,000 times, and calculated the $p$ value of $S$ as the ratio of $S' \geqq S$.

### Habitat preference analysis of Lactobacillaceae based on metagenome
To identify species inhabiting fructose-rich environments, we estimated the habitat preference for each of the 336 Lactobacillaceae species for which high-quality representative genomes and 16S rRNA sequences were available from GTDB (see Datasets). We queried the 16S rRNA sequences of the 336 species against ProkAtlas online[62] on June 30, 2021. ProkAtlas conducts a BLAST search of queried 16S rRNA sequences in short-read metagenomic resources from various environments (e.g., soil, human gut, and pollen), then returns a habitat preference score for each environment. ProkAtlas parameters for nucleotide identity and sequence coverage thresholds were set to 97% and 150 bp, respectively.

### Detection of *adhE*-like genes

*adhE* genes possessed by Lactobacillaceae were comprehensively detected, as shown in Supplementary Fig. 6a. We firstly searched *adhE* (K04072 in the KEGG Orthology database[48]) for all the 25,877 high-quality representative genomes of all bacterial phyla by Hidden Markov Model-based ortholog annotation using KofamScan v1.3.0 (ref. 47). Based on the OG annotation results, we extracted all genes annotated as *adhE* and filtered genes encoding proteins in the length range between 800-1000 aa, which yielded 4833 genes. We then conducted a sensitive homology search for every protein sequence of the 369 Lactobacillaceae species, treating the 4833 genes as a database using MMseqs2 13.45111 (mmseqs easy-search -s 7.50)[60]. By extracting genes with both target coverage and identity greater than 45% for any of the 4833 genes, we retrieved 399 *adhE*-like genes from Lactobacillaceae.

### Classification of *adhE*-like genes into *adhE*-derived genes and other gene families

To remove genes belonging to different families from the 399 *adhE*-like genes, we constructed phylogenetic trees for both the N- and C-terminal regions of AdhE. We first conducted a multiple sequence alignment (MSA) for the 399 *adhE*-like genes in Lactobacillaceae and the *adhE* gene of *Escherichia coli* (here assumed to be an outgroup within the AdhE family) using MAFFT 7.310 (ref. 77). We then extracted 500 N-terminal and C-terminal columns from the resulting MSA and excluded sequences whose >50% of the columns were filled with gaps in the extracted MSAs, which yielded MSAs of 381 and 314 sequences for the N-terminal and C-terminal half regions of *adhE*, respectively. After trimming MSA columns filled with gaps by Trimal v1.3. rev15 (-gappyout)[78], phylogenetic trees were constructed for both MSAs using IQ-TREE v2.0.3 (-m MFP -bb 1000 -nt 20)[79]. For the two phylogenetic trees, we first estimated the latest common ancestor node on the phylogenies by assuming *E. coli adhE* was an outgroup of all Lactobacillaceae complete *adhE* genes, which are ~900 aa. Finally, we identified an outgroup clade of all *adhE*-like genes and treated them as non-*adhE* genes, that is, genes belonging to other protein families. After excluding these sequences, we obtained 372 sequences of *adhE* and *adhE*-derived fragment genes possessed by Lactobacillaceae.

To verify the validity of the non-*adhE* gene identification, we constructed sequence similarity networks (SSNs) for 399 adhE-like genes. We conducted an all-versus-all alignment of the sequences by MMseqs2 (easy-search -s 7.50) and constructed two SSNs by linking all gene pairs with >45% and >70% sequence identity, respectively.

### *adhE* detection in representative and non-representative genomes of *Companilactobacillus zhachilii*, *Lacticaseibacillus saniviri*, and *Lentilactobacillus kefiri*

Representative/non-representative genomes of the three species in GTDB were downloaded by ncbi-genome-download-0.3.1 from Refseq and Genbank, and we annotated genes in the genomes by prodigal 2.6.3 (ref. 80). We then searched *for adhE*-like genes in these genomes using the same method used to detect *adhE*-like genes in representative Lactobacillaceae genomes. Finally, we aligned the detected genes using MAFFT and visualized the multiple sequence alignments.

### Rate parameter estimation for state transitions of AdhE repertoire

To analyze the evolutionary dynamics of AdhE repertoire, we counted AdhEs per species to distinguish complete AdhEs, ALDH fragments, ADH fragments, and intermediates (partially missing ALDH or ADH domain) based on the sequence length and center position of every sequence in the MSA. Here, complete AdhEs, ALDH fragments, ADH fragments, and intermediates were defined as sequences with >800 aa lengths and center positions in the 400-900th columns of MSA, those with center positions in the <400th columns, those with center positions in >900th columns, and the other sequences.

To estimate the state transition rates among the seven observed states of the AdhE repertoire per species (color-filled circles in Fig. 3d), we

conducted a phylogenetic comparative analysis using the Mk model (k = 7) implemented in Diversitree package. All state transition parameters (42 parameters) were estimated using the maximum likelihood method (method = "nlm"), given as inputs the species tree of 369 Lactobacillus species and the state information of the AdhE repertoire for each species. Each parameter was initialized to 1/42.

### Analysis of mutations accumulating specifically in *adhE*-derived fragment genes

To detect mutations accumulating specifically in *adhE*-derived fragment genes, we conducted ancestral sequence reconstruction of the N-terminal half regions of 381 *adhE*-like genes using IQ-TREE v2.0.3 (-m MFP -asr -te). The tree topology was fixed as described above. We adopted one amino acid with a > 50% posterior probability for each sequence position of every ancestral node. If no amino acid showed >50% posterior probability, the site at the node was treated as uncertain. For every sequence position in the MSA, we detected branches of the phylogenetic tree where one or more substitutions occurred based on whether different amino acids were adopted for the branch's parental and child nodes. By counting the branches where substitutions occurred, we detected the top-15 sequence positions with substitutions specific to the clades of *adhE*-derived N-terminal fragments.

### Structural analysis of AdhE

To interpret the functional insights into mutations at 15 sequence positions, we downloaded the structure of the AdhE complex (6ahc) from Protein Data Bank, extracted a dimer structure (chains B and C), and mapped the positions onto it. Next, we tested the enrichment of these positions at the interaction interface of AdhEs or at the active site of the AdhE ALDH domain. We annotated 88 residues of chain B, whose any atom is within 5Å of chain C, as residues for dimerization of AdhE. We also downloaded and analyzed another structure of AdhE bound to an $NAD^+$ (6tqh[66]) and defined 41 residues of chain B, whose any atom is within 5Å of the $NAD^+$ molecule, as residues for the active site of the ALDH domain. We then compared the portion of substitutions in the ALDH fragment clades among residues for dimerization, the active site of the ALDH domain, and other residues.

### Statistics and reproducibility

Throughout the paper, we selected appropriate statistical tests for each format of analyzed data. We selected Mann–Whitney $U$-test or Phylogenetic ANOVA to test the significance of differences between data groups when the data are independent or dependent from a phylogeny, respectively. We chose Chi-squared test for testing significant differences between expected and observed values of contingency tables. Furthermore, we conducted permutation tests on the significance of gene loss order similarity. All analyses in this study are reproducible using public datasets or the source data of this study (Supplementary Data 4).

### Data availability

The bacterial reference phylogeny and genome sequence data used in this study are publicly available from GTDB. We provide the list of genome accessions for Lactobacillaceae species for which we analyzed their genomes and habitats (Supplementary Data 2). In addition, the list of 137 orthologs commonly and independently lost in two FLAB lineages is provided (Supplementary Data 3). Amino acid sequences of *adhE*-like genes are provided as Supplementary Data 1. The source data behind figures in this paper are provided as Supplementary Data 4. These supplementary materials are available in a public repository, Zenodo (https://doi.org/10.5281/zenodo.11378208)[81]. All data needed to evaluate the conclusions in the paper are present in the paper, the Supplementary Materials, and/or public repositories described above.

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

## Acknowledgments

We thank all the members of the Iwasaki lab for valuable discussions and critical comments on the content of this paper, especially T. K. Suzuki and K. Miyake for many insightful feedbacks. Computations were partially performed on the SuperComputer System, Institute for Chemical Research, Kyoto University. This work was funded by JPMJCR19S2 from Japan Science and Technology Agency to W.I.; KAKENHI 19H05688 and 22H04925 from the Japan Society for the Promotion of Science (JSPS) to W.I.; and KAKENHI 22J20318 from the JSPS to N.K.

## Author contributions

Conceptualization: N.K., S.M., A.E., and W.I. Methodology: N.K., S.M., A.E., and W.I., Investigation: N.K., S.M., and Y.T., Visualization: N.K., Supervision: W.I., Writing—original draft: N.K., A.E., and W.I., Writing—review and editing: N.K., Y.T., M.A., A.E., and W.I.

## Competing interests

The authors declare no competing interests.
