## [Peer Review File · Communications Biology]

Reviewers' comments:

Reviewer #2 (Remarks to the Author):

The manuscript describes bioinformatic analyses to inform on the evolution of the genera *Fructobacillus* and *Apilactobacillus*, focusing on the two-domain acetaldehyde dehydrogenase / alcohol dehydrogenase AdhE, which it found in the genomes of most heterofermentative lactobacilli but not in “fructophilic” lactobacilli, which thus lack the ability to convert acetyl-phosphate to ethanol and require fructose as electron acceptor for growth. The manuscript informs that in addition to the lack of AdhE, fructophilic lactobacilli share several other genomic features, particularly gene loss. Overall, the analyses described improve our understanding of the evolution and ecology of lactobacilli. The manuscript has several shortcomings and inconsistencies, however, that should be addressed in a manuscript revision.

1) Figure 1B appears to show a gene content tree, which has the homofermentative genus *Pediococcus* clustering with some of the heterofermentative lactobacilli. Most or all of the recent core genome phylogenetic trees including Figure 4B, however, document a single node separating homofermentative from heterofermentative lactobacilli. Gene gain / loss of specific clades / genera is preferably shown with a phylogenetic tree (see e.g. Figure 3 of <http://dx.doi.org/10.1128/AEM.02116-15>).

2) Figures 2E and 2F inform that the majority of genes were lost with the transition from a free-living lifestyle to an insect associated lifestyle (*Convivina* and *Fructobacillus* as well as *Fructilactobacillus* and *Apilactobacillus* (for the latter two also shown in Figure 3 of <http://dx.doi.org/10.1128/AEM.02116-15>). In addition, several genera that share major genomic, physiological and metabolic features with apilactobacilli were not included in the analysis (*Acetilactobacillus*, *Philodulcिलactobacillus*, *Nicoliella*). Inclusion of these organisms is likely to strengthen conclusions on genomic features of the transition to insect adapted microbes.

3) The selection of genomes that were used for the genome analyses is not justified. Not all genera / species of the Lactobacillaceae are represented but few species are represented by several genomes.

4) Figure 3 informs on the gene categories that were lost in addition to AdhE. Specific genes that were lost with the transition to fructophilic LAB and / or from a free living to an insect associated lifestyle should be indicated in more detail.

5) Several past studies inform that the two domain AdhE is absent in a vast majority of the type strain genomes of homofermentative lactobacilli (e.g. Figure 4 of <http://dx.doi.org/10.1128/AEM.02116-15>). The discrepancy to the analyses presented here should be discussed, in particular with reference to the BLAST query sequence that was used to retrieve AdhE in the genome dataset, and with respect to BLAST parameters. The corresponding

methods section (lines 484 to 490) informs that 4833 AdhE genes were identified in genomes of Lactobacillaceae but does not inform on the total number of (homofermentative / heterofermentative) genomes of the Lactobacillaceae. In addition, a query coverage of 45% was used – this approach is likely flawed as a higher coverage should be used to identify the > 800 aa two domain enzyme. The parameters 45% coverage / 45% aa identity possibly or likely identify alcohol dehydrogenases that differ in structure and function from the two-domain AdhE. (It is noted that BLAST analysis with AdhE of *Fl. sanfranciscensis* or *Lm. fermentum*) against all Lactobacillaceae with 75% as cut-off still retrieves many sequences of homofermentative lactobacilli so the proportion of positive / negative genomes per genus / clade may be important.

The presence of the two domain AdhE in homofermentative lactobacilli should be discussed in relation to pyruvate conversion to pyruvate formate lyase. Comparable to the phosphoketolase pathway in heterofermentative lactobacilli, this pathway require conversion of acetyl phosphate to ethanol.

Methods:

- line 434. Genomes of which phyla? To understand the evolution of fructophilic lactobacilli, it is sufficient to look only at Lactobacillaceae?
- line 434. How many of the 45,555 were Lactobacillaceae? How were the 369 genomes selected that were used to build the phylogenetic tree?
- the 4833 AdhE encoding genomes of the Lactobacillaceae represent which proportion of the total number of genomes? Was the proportion different in homo- and heterofermentative lactobacilli?

Reviewer #3 (Remarks to the Author):

In this manuscript, Kondo et al propose that convergent evolution in two bacterial lineages that have independently adapted to a fructose-rich diet provide evidence for a deterministic evolutionary process both on the gene set level and on the level of protein domain architecture evolution. Overall, I consider it to be an interesting idea to challenge the idea of a stochastic evolutionary process with the concept of determinism. Unfortunately, I do not think that the data and analyses they describe here, though certainly very interesting, allow to conclude on the existence of a deterministic evolutionary process. Instead, they point towards non-obvious functional connections between the convergently lost genes that allow only a certain order of loss events during the adaptation process to maintain a sufficiently fit evolutionary lineage. To give one example, a metabolic pathway that generates, as an intermediate product, a substance that is toxic in higher concentrations cannot degrade from bottom to top, where top is the start of the reaction chain. Instead, the pathway must degrade during evolution in a way that an

accumulation of potentially toxic products must be avoided. Of course, the evolutionary process is unaware of this, and thus mutations that result in an accumulation of the toxic substance do occur, but they are lost from the population very quickly. I think that the data in this manuscript would be excellently suited to unravel such dependencies.

Major issues

1. I have the impression that the entire abstract benefits from a thorough overhaul that makes the relevant information to better stand out.
 - a. I do not think that there is any 'necessity' in evolution, since evolution simply refers to change over time. When the authors talk about 'necessity' it might be better to introduce this term in the context of (evolutionary) adaptation. Some of the changes seen between species are due to chance, others reflect the 'necessity' to adapt to a particular ecological niche.
 - b. Line 31. The sentence starting with 'Although' requires revision
 - c. Line 34. I don't think that the word construct 'phylogenetic comparative genomics' is particularly helpful. I suggest to stick with comparative genomics (or comparative gene set analyses) then mentioning that similar patterns of gene loss have been observed on independent evolutionary lineages.
 - d. Lines 34-35. It seems that his sentence provides information that is then repeated in the next sentence. If here it is already clear that different LAB clades dwelling in the same ecological niche show a 'convergent' gene repertoire, then the information in the next sentence becomes irrelevant.
 - e. Line 37. The phrasing "We revealed that (...) tended to follow (...)" is sub-optimal. To my understanding, 'revealing' should be used if a clear signal was detected. 'tended to follow', however, is anything but a clear signal. Moreover, the authors should consider that they use 'reveal' again in the next sentence.
2. Similar to the above issue, the entire manuscript would benefit from a very thorough check of the write up with the aim to improve precision and clarity.
3. In the introduction, the authors state "convergent evolution is a powerful clue for revealing the deterministic nature of evolution". As outlined above, I do understand the intention behind this statement, which is ok in principle. Still, I have to strongly object when it comes to the wording. Determinism is defined as "events are completely determined by previously existing causes". Evolution, as the heritable change of genetic information, however is a chance process without any deterministic component. I have the impression that the authors confuse determinism with the outcome of a strong selective pressure that selects from a randomly generated set of variants only a very specific subset whereas all other variants go extinct. This is latter aspect is then, Indeed, covered by the second part of the sentence, where the authors write "independent but common evolutionary outcomes strongly suggest the existence of shared selective pressures or

constraints.” I therefore strongly encourage the authors to carefully re-think the use of the term ‘Determinism’. In line with this, their evolutionary stream model should be better seen as one realization of an ‘evolutionary funnel’ process. From all possible realizations, a stream could be preferentially selected when success, i.e., the adaptation to an ecological niche, is granted only when events happen (by chance) in the correct order. If the authors indeed want to stick with ‘determinism’, then they have to show that in a population of independently evolving individuals a mutation event A always precedes a mutation event B, and if event A has happened then only event B and not event C can happen. The authors may further consider that Giannakis et al. 2022 address the question what ‘determines’ which genes are retained in organelles. The rules they propose reflect the selection process and not the mutation process.

4. In their analysis, the authors use different 500 different parameter settings to infer the ancestral state for their orthologous groups (OG). They then assign a binary vector of length 3 to each OG, representing the presence of the corresponding gene at three internal nodes according to each of the 500 replicates. They then state that they selected the most strongly supported vector across the replicates to represent the OG. I think what is missing here is an information about the dependencies between the parameter values used for the ancestral state reconstruction. If a subset of the analyses were performed with values drawn from a certain region in the parameter hyperspace, then the outcome of the inferences will be likely similar. Moreover, I interpret Fig. 2c such that the results of the ancestral state reconstruction remain largely unchanged when modifying the parameter values. How does this reconcile with the statement in the main text “Because the estimation of ancestral gene content was uncertain(...)”, which suggests that results change when parameter values are modified?

5. Bacteria are renowned for sharing genes via recombination and HGT. Such events are not considered here. Moreover, many of the metabolic genes are organized in functional gene clusters. How does this influence the outcome of this analysis?

6. Lines 157-159. I do not follow the argumentation. However, this seems important for the study. Please explain and rephrase

7. Fig. 3 a represents a standard permutation test to come up with an empirical distribution how similarities of two loss orders are distributed by chance. I do not think that the manuscript benefits much from this figure, and they could as well be placed into the supplement.

8. The search for *adhE* homologs is confusing. Why is it that the authors did not use standard ortholog searches to determine the phylogenetic profile of this gene? What is the rationale to first search for *adhE* like proteins in diverse bacterial phyla returning then to their focal organisms to perform the ortholog search? Moreover, it should be readily accessible if this search was done indeed on the gene level, as suggested by the text, or by analysing the predicted protein sets. When it comes to establishing phylogenetic profiles of proteins across large taxon collections keeping track of changes in ortholog length and feature architecture differences, the authors may want to take a look at [10.3389/fmicb.2021.739000](https://doi.org/10.3389/fmicb.2021.739000) and

10.1371/journal.pgen.1010646

9. The state transition model shown in Fig. 4f is interesting. However, it has to be way better integrated into the story in order to provide a relevant contribution. Moreover, many of the transitions in the graph remain unexplained. Looking closely, it appears as if it would allow transitions that require two independent changes. Is this intentional? If so, then this has to be motivated and explained. On top of this, the model has a transition from 'no adhE-derived' gene to 'complete x 1'. What kind of event is this? A lateral acquisition of a gene? Lastly, why is it that the authors talk about an adhE-derived gene' instead of adhE?

10. I am missing a functional intuition why certain bacterial lineages have a high preference of retaining the ALDH domain rather than the ADH domain. I would be surprised if there is no considerably simple explanation. Naively, I would assume that an orphan ALDH domain can be, with considerably few adaptations be re-used in a different functional context. This might not be possible for the ADH domain.

Minor issues

1. Figure 1. Please correct 'Sifferent' to 'Different'
2. Line 77. The authors may want to revise their statement that FLABs grow poorly on fructose
3. the more appropriate reference of the absence of AdhE in FLAB might be 10.1007/s00284-013-0506-3
4. Line 119. The authors may want to comment on the selection of a 'representative' genome for each of the 344 species, and how the selection of a different representative may change conclusions.
5. Figure 2a. The third taxon from the bottom displays an ortholog for 'b'. In the ancestral state reconstruction, this is ignored. What is the rationale here? Is this ortholog considered to be a xenolog instead, and based on what evidence?
6. Line 116. How many orthologous groups were analysed? This info is given in Fig 2C but not in the main text.
7. Line 124. The authors may want to use LCA instead of CA, just for the sake of precision
8. Line 130. The wording 'robust among' should be revised
9. Line 151. Consider rephrasing to 'larger than expected by chance'. Either wording, however, requires the result of a statistical test
10. Line 154. What is the difference between a 'pressure' and a 'constraint'? Moreover, either word likely requires the addition of 'selective'
11. Line 156. Please rephrase this sentence. Genes cannot converge in this analysis. It is the composition of gene sets that can converge.
12. Line 226. Please introduce GTDB upon its first use
13. Line 228. Please explain how you used 4,833 adhE like genes as query for homology searches. This should be already understandable from reading the main text

14. Line 230. Please correct to 'substantially shorter'
15. Line 237. Please revise sentence. As it stands, it is hard to grasp.
16. Line 240. The shotgun metagenomics analysis comes out of nowhere and needs a better integration.
17. Line 248. I think it is not appropriate to say that two taxa 'share evolutionary pressure'. Rather they are subjected to the same selective pressure
18. Line 281. The word 'rigorous' generates a wrong impression. Instead, using 'sensitive' would be a better option.
19. Line 295. If the authors want to make a claim that ALDH-only AdhE proteins are a major evolutionary intermediate, then Fig4d is not optimal to support this claim. Instead, it would be necessary to show that an ancestral state reconstruction reveals a fragmented AdhE at the internal nodes connecting species that have either lost AdhE or still retain a truncated version.
20. Line 311. What is the relevance of the half sentence 'or common domain-loss order'?
21. Line 325. It is better to replace 'fixed' which is a term from population genetics by 'invariable' or 'conserved'. Moreover, the argumentation here appears circular. 15 sites are selected because they vary across the ALDH-only fragments but are conserved in the complete adhE genes. In the next sentence, the authors then state as a result that amino acid composition indeed shows that they are less conserved. I regret that I cannot see the relevance of this statement.

I hope that my comments help in improving the manuscript.

Reviewer #4 (Remarks to the Author):

Konno et al. present a comprehensive phylogenetic analysis of a large number of lactic acid bacteria to determine whether convergent evolution follows stochastic or deterministic paths. They showed that distinct lineages of fructophilic lactic acid bacteria have experienced shared losses of orthologs and gene loss. Moreover, they have also identified that a loss of gene adhE led to the convergent evolution of FLAB which followed a specific evolutionary path in multiple lineages of lactic acid bacteria sharing fructose rich habitats. This work brings new insights into convergent evolution across multiple biological levels, ranging from amino acid to a protein domain. I believe this work will be of interest to the researchers working in the field of microbial evolution and phylogenetics. Overall, the manuscript is convincing: experiments are well designed and analyzed, the level of detail in the 'Methods' section is appropriate, and the conclusions are supported by the data.

While I'm familiar with some of the experiments performed in this manuscript, I don't have much experience handling large-scale phylogeny data. So please forgive my lack of clarity if it comes up in places.

The idea that long-term evolutionary processes share evolutionary paths at multiple levels is quite an interesting topic for investigation. However, I felt that the idea of convergent evolution needs to be explained much more clearly. For example, how and why convergent evolution is deterministic? Can you define the separate evolutionary events leading to multiple lineages? (authors could cite these references – Stayton, T. 2015 and Stern, D. 2013)

I believe the model presented here needs further explanation, clear supporting evidence from previous studies and how these evolutionary models actually test your hypothesis. Authors have listed a few studies that support the presented evolutionary models. However, the rationale behind adopting these two alternating strategies remains unclear.

Line 84: The text explaining the result needs further clarification. This seems like an important result, but it was hard to understand the explanation of how the order of gene loss supports the deterministic evolutionary stream model.

Please find more precise comments below:

1) Line 40: Are authors trying to establish FLAB as a model system?

I found it very difficult to extract any meaning from this sentence.

2) Line 44: "both" is misleading in the sentence since the authors talk about three key aspects driving the evolution.

3) Line 46: The statement "Convergent evolution is a powerful clue for revealing the deterministic nature of evolution because independent but common evolutionary outcomes strongly suggest the existence of shared selective pressures or constraints" needs citation.

4) Line 48: The sentence talks about previous studies, but those studies haven't been mentioned either as a citation or as an example.

5) Fig. 1 Figure panels should have subheadings a) Evolutionary funnel model b) Evolutionary stream model

"...Sifferent evolutionary paths" Is this a typo? I guess it has to be "different evolutionary paths"

6) Line 76: The sentence is misleading and contrary to the definition of FLAB (Endo et al 2018). Can they really grow poorly on fructose?

7) Line 79: I found it very hard to extract any meaning from this sentence. I think there is too much information packed in a single sentence. I recommend you unpack it into a few different sentences so that you can clearly express these ideas.

8) Line 100-107: These are results and should not be included in the introduction.

- 9) Fig. 2a: You need to specify what does a and b orthologs represent.
- 10) Fig.2d: does this contingency table need statistical analysis?
- 11) Fig.3b: Can you please specify what is the significant p value threshold?
- 12) Fig.3b: What does "ALL" signify?
- 13) Fig.3d: It is unclear what the size of the dot indicates here. It would be good to mention this in the figure legend.
- 14) Line 232: This sentence is hard to understand. I would recommend breaking it into two separate sentences.
- 15) Line 388: I believe the use of the word "Instead" is redundant here.

Reviewer #2 (Remarks to the Author):

The manuscript describes bioinformatic analyses to inform on the evolution of the genera *Fructobacillus* and *Apilactobacillus*, focusing on the two-domain acetaldehyde dehydrogenase / alcohol dehydrogenase *AdhE*, which it found in the genomes of most heterofermentative lactobacilli but not in fructophilic lactobacilli, which thus lack the ability to convert acetyl-phosphate to ethanol and require fructose as electron acceptor for growth. The manuscript informs that in addition to the lack of *AdhE*, fructophilic lactobacilli share several other genomic features, particularly gene loss. Overall, the analyses described improve our understanding of the evolution and ecology of lactobacilli. The manuscript has several shortcomings and inconsistencies, however, that should be addressed in a manuscript revision.

We deeply appreciate all of your comments and suggestions. Here we have carefully revised and improved our manuscript thanks to your questions and comments, especially reconsidering the homofermentative species missing *adhE* and the reference tree topology of Lactobacillaceae. We hope you will be satisfied with the revised version of our manuscript.

1) Figure 1B appears to show a gene content tree, which has the homofermentative genus *Pediococcus* clustering with some of the heterofermentative lactobacilli. Most or all of the recent core genome phylogenetic trees including Figure 4B, however, document a single node separating homofermentative from heterofermentative lactobacilli. Gene gain / loss of specific clades / genera is preferably shown with a phylogenetic tree (see e.g. Figure 3 of <http://dx.doi.org/10.1128/AEM.02116-15>).

Thank you for your suggestion. For clarity, the phylogeny in both Figure 2b and 4a is not a gene-content tree but is a core genome phylogenetic tree provided in the GTDB database. Thanks to your comment, we noticed that the phylogenetic location of *Pediococcus* is different between that GTDB-based phylogeny and another reported Lactobacillaceae phylogeny you introduced (PMID: 26253671). We carefully validated the robustness of the GTDB-based tree topology by re-inferring a genomic phylogeny (L237-239). Furthermore, we also tested the robustness of our result by regrafting or removing the *Pediococcus* clade in the original GTDB-based phylogeny (L243-246). As you suggested, we also visualized gene gain/loss numbers on each branch of the reference phylogeny by referring to the previous study (PMID: 26253671) (L202-204; **Supplemental Fig. 3**).

2) Figures 2E and 2F inform that the majority of genes were lost with the transition from a free-living lifestyle to an insect associated lifestyle (*Convivina* and *Fructobacillus* as well as *Fructilactobacillus* and *Apilactobacillus* (for the latter two also shown in Figure 3 of <http://dx.doi.org/10.1128/AEM.02116-15>). In addition, several genera that share major genomic, physiological and metabolic features with apilactobacilli were not included in the analysis (*Acetilactobacillus*, *Philodulcिलactobacillus*, *Nicoliella*). Inclusion of these organisms is likely to strengthen conclusions on genomic features of the transition to insect adapted microbes.

We appreciate this comment. As reviewer 3 also asked about the species selection in this study, we enriched the explanation of our taxon sampling criteria. In the present study, we used all the available high-quality representative genomes in GTDB r202. We set >95% completeness and <5% contamination as the criteria of high-quality genomes because our gene-content analysis relies on the accuracy of gene presence/absence information. We also tried to include genomes of the phylogenetically related clades to FLAB (i.e., *Acetilactobacillus*, *Philodulcिलactobacillus*, and *Nicoliella*), but even the latest version of GTDB (r214) did not contain genomes satisfying our genome quality criteria in those clades. We clarified this point in our main manuscript (L136-140).

3) The selection of genomes that were used for the genome analyses is not justified. Not all genera / species of the Lactobacillaceae are represented but few species are represented by several genomes.

We also appreciate this comment. As we mentioned as the answer to your comment 2, we used all the available high-quality (>95% completeness and <5% contamination) representative genomes in GTDB r202.

4) Figure 3 informs on the gene categories that were lost in addition to AdhE. Specific genes that were lost with the transition to fructophilic LAB and / or from a free living to an insect associated lifestyle should be indicated in more detail.

Thank you for your suggestion. To indicate FLAB-specific gene losses in detail, we provided the list of 137 KEGG orthologs commonly lost in the FLAB lineages and the information of branches where each ortholog was lost in **Supplemental Table 2**.

5) Several past studies inform that the two domain AdhE is absent in a vast majority of the type strain genomes of homofermentative lactobacilli (e.g. Figure 4 of <http://dx.doi.org/10.1128/AEM.02116-15>). The discrepancy to the analyses presented here should be discussed, in particular with reference to the BLAST query sequence that was used to retrieve AdhE in the genome dataset, and with respect to BLAST parameters. The corresponding methods section (lines 484 to 490) informs that 4833 AdhE genes were identified in genomes of Lactobacillaceae but does not inform on the total number of (homofermentative / heterofermentative) genomes of the Lactobacillaceae. In addition, a query coverage of 45% was used. This approach is likely flawed as a higher coverage should be used to identify the > 800 aa two domain enzyme. The parameters 45% coverage / 45% aa identity possibly or likely identify alcohol dehydrogenases that differ in structure and function from the two-domain AdhE. (It is noted that BLAST analysis with AdhE of *Fl. sanfranciscensis* or *Lm. fermentum*) against all Lactobacillaceae with 75% as cut-off still retrieves many sequences of homofermentative lactobacilli so the proportion of positive / negative genomes per genus / clade may be important.

Thank you for your comments; Reviewer 3 also asked about this point. Overall, our analysis framework was made to improve the sensitivity of *adhE* detection compared to the previous study you introduced (PMID: 33563209), where only a sequence similarity search by BLASTp was conducted. For clarity, the 4,833 *adhE* genes were possessed not only by Lactobacillaceae genomes but by 25,877 genomes of all the bacterial phyla. We detected the 4,833 *adhE* genes by Hidden Markov Model (HMM)-based sequence search against all phyla of bacteria by KofamScan and excluding hits shorter than 800 amino acids. We set the thresholds of the sequence similarity search against Lactobacillaceae genomes as 45% coverage and 45% aa identity to sensitively detect previously reported partial *adhE* genes as well as complete *adhE* genes (PMID: 33563209).

At the same time, as you mentioned, we were noted that the low thresholds of the coverage and the identity can cause misdetection of other ALDH-only or ADH-only orthologs. Therefore, we conducted phylogenetic analyses for the detected 399 *adhE*-like genes to exclude ALDH-only or ADH-only non-*adhE* orthologs. We identified 27 non-*adhE* genes in the 399 genes and excluded them from the following analyses. We elaborated on the explanation of this point in L267-282. We also confirmed the detection of non-*adhE* genes is supported by sequence similarity network analyses as discussed in L341-342.

6) The presence of the two domain AdhE in homofermentative lactobacilli should be discussed in relation to pyruvate conversion to pyruvate formate lyase. Comparable to the phosphoketolase pathway in heterofermentative lactobacilli, this pathway require conversion of acetyl phosphate to ethanol.

We deeply appreciate this comment. As you suggested, the reaction catalyzed by pyruvate formate lyases in homofermentative lactobacilli can be coupled with AdhE-catalyzed reactions. To test the hypothesis, we visualized the phylogenetic distribution of pyruvate formate lyase (KEGG Ortholog identifier: K00656) and compared it with that of *adhE*. Surprisingly, pyruvate formate lyases are conserved only in homofermentative lactobacilli, and homofermentative species missing pyruvate formate lyases tended to miss *adhE*. This suggests that the pyruvate formate lyase and *adhE* are functionally coupled and tend to be lost together during evolution. We added this finding in our main manuscript with a supplemental figure (L287-291, **Supplemental Figure 6f**).

Methods:

7) line 434. Genomes of which phyla? To understand the evolution of fructophilic lactobacilli, it is sufficient to look only at Lactobacillaceae?

In the present study, we used representative genomes and phylogenies only of Lactobacillaceae throughout this study, except for the detection of the 4,833 complete *adhE* genes from 25,877 genomes of all bacterial phyla. We clarified this point in L512-513. The rationale behind analyzing only Lactobacillaceae genomes was that it contains the latest common ancestor of two fructophilic lactobacilli in the reference species phylogeny.

8) line 434. How many of the 45,555 were Lactobacillaceae? How were the 369 genomes selected that were used to build the phylogenetic tree?

Thank you for asking this. The 45,555 bacterial species included 369 Lactobacillaceae species and we analyzed all of the genomes to detect *adhE* genes. For the gene-content analysis, we also extracted a tree of Lactobacillaceae for 344 species with high-quality representative genomes (defined as >95% completeness and <5% contamination throughout this study). We clarified this point in L505-512.

9) the 4833 AdhE encoding genomes of the Lactobacillaceae represent which proportion of the total number of genomes? Was the proportion different in homo- and heterofermentative lactobacilli?

Thank you for asking this question, too. For clarity, the 4,833 *adhE* genes are not only of Lactobacillaceae species but of species in all the bacterial phyla. Within 369 Lactobacillaceae species, we detected 372 *adhE* genes from 324 species in total. We detected *adhE* in 84.3% and 94.0 % of analyzed homofermentative and heterofermentative species in Fig. 4a, respectively.

Reviewer #3 (Remarks to the Author):

In this manuscript, Kondo et al propose that convergent evolution in two bacterial lineages that have independently adapted to a fructose-rich diet provide evidence for a deterministic evolutionary process both on the gene set level and on the level of protein domain architecture evolution. Overall, I consider it to be an interesting idea to challenge the idea of a stochastic evolutionary process with the concept of determinism. Unfortunately, I do not think that the data and analyses they describe here, though certainly very interesting, allow to conclude on the existence of a deterministic evolutionary process. Instead, they point towards non-obvious functional connections between the convergently lost genes that allow only a certain order of loss events during the adaptation process to maintain a sufficiently fit evolutionary lineage. To give one example, a metabolic pathway that generates, as an intermediate product, a substance that is toxic in higher concentrations cannot degrade from bottom to top, where top is the start of the reaction chain. Instead, the pathway must degrade during evolution in a way that an accumulation of potentially toxic products must be avoided. Of course, the evolutionary process is unaware of this, and thus mutations that result in an accumulation of the toxic substance do occur, but they are lost from the population very quickly. I think that the data in this manuscript would be excellently suited to unravel such dependencies.

We deeply thank all of your comments and suggestions. Here we thoroughly revised and improved our manuscript thanks to your questions and comments. In particular, we enriched Introduction to explain our ideas aligned with previous studies, carefully checking if the choices of words were appropriate to describe what we studied. We also elaborated the explanation of the computational pipeline to detect *adhE*. We hope you will enjoy the revised version of our manuscript.

Major issues

1. I have the impression that the entire abstract benefits from a thorough overhaul that makes the relevant information to better stand out.

a. I do not think that there is any “necessity” in evolution, since evolution simply refers to change over time. When the authors talk about “necessity” it might be better to introduce this term in the context of (evolutionary) adaptation. Some of the changes seen between species are due to chance, others reflect the “necessity” to adapt to a particular ecological niche.

We deeply thank you for this comment. Whereas the term “necessity” has been used to describe evolutionary processes including the gene-loss evolution (PMID: 11234024; 16572170; 34061027), we avoided using the term “necessity” in the revised manuscript including the abstract in our revised version. We used the word “non-random” to describe independent evolutionary processes showing higher similarity than expected by chance (L30).

b. Line 31. The sentence starting with “Although” requires revision

Thank you so much. We revised the sentence to correct the grammar. Notably, we thoroughly revised the Abstract and removed the term “deterministic” for the same reason as the reason we avoided using the term “necessity.” Instead, we wrote “it remains unclear whether macroevolutionary convergence follows stochastic or constrained paths” (L32-33). We hope you can approve our revised version of the Abstract.

c. Line 34. I don't think that the word construct “phylogenetic comparative genomics” is particularly helpful. I suggest to stick with comparative genomics (or comparative gene set analyses) then mentioning that similar patterns of gene loss have been observed on independent evolutionary lineages.

We agree with your suggestion. We used the terms “comparative genomics” and “phylogenetic comparative methodologies” to describe our analysis framework in the revised version of Abstract.

d. Lines 34-35. It seems that this sentence provides information that is then repeated in the next sentence. If here it is already clear that different LAB clades dwelling in the same ecological niche show a “convergent” gene repertoire, then the information in the next sentence becomes irrelevant.

We appreciate your comment. While the convergence of gene repertoire was previously suggested (PMID: 33563209), we newly suggest that the convergence was caused by gene losses of 137 orthologs, and the gene losses occurred in a significantly similar order by phylogenetic comparative methods. We clarified this point in the revised version of Abstract (L36-38).

e. Line 37. The phrasing “We revealed that tended to follow” is sub-optimal. To my understanding, “revealing” should be used if a clear signal was detected. “tended to follow”, however, is anything but a clear signal. Moreover, the authors should consider that they use “reveal” again in the next sentence.

We agree with your point. We replaced the word “revealed” in the phrase, “We revealed that tended to follow,” with “suggested”. (L39)

2. Similar to the above issue, the entire manuscript would benefit from a very thorough check of the write up with the aim to improve precision and clarity.

We would like to thank all the above comments, and we thoroughly checked the words and phrases to revise our manuscript.

3. In the introduction, the authors state “convergent evolution is a powerful clue for revealing the deterministic nature of evolution”. As outlined above, I do understand the intention behind this statement, which is ok in principle. Still, I have to strongly object when it comes to the wording. **Determinism is defined as “events are completely determined by previously existing causes”**. Evolution, as the heritable change of genetic information, however is a chance process without any deterministic component. I have the impression that the authors confuse determinism with the outcome of a strong selective pressure that selects from a randomly generated set of variants only a very specific subset whereas all other variants go extinct. This is latter aspect.

Indeed, covered by the second part of the sentence, where the authors write “independent but common evolutionary outcomes strongly suggest the existence of shared selective pressures or constraints.” I therefore strongly encourage the authors to carefully re-think the use of the term “Determinism”. In line with this, their evolutionary stream model should be better seen as one realization of an “evolutionary funnel” process. From all possible realizations, a stream could be preferentially selected when success, i.e., the adaptation to an ecological niche, is granted only when events happen (by chance) in the correct order. If the authors indeed want to stick with “determinism”, then they have to show that in a population of independently evolving individuals a mutation event A always precedes a mutation event B, and if event A has happened then only event B and not event C can happen. The authors may further consider that Giannakis et al. 2022 address the question what “determines” which genes are retained in organelles. The rules they propose reflect the selection process and not the mutation process.

We deeply appreciate this comment and thank you so much for clarifying your thoughts. We have avoided using the term

“deterministic” and, for example, used the word “non-random” to describe evolutionary processes that cannot be fully explained by chance. For clarity, we defined the evolutionary stream and funnel models as the models of evolution where intermediate steps (e.g., loss of multiple genes) toward convergence occurred in a random and constrained order, respectively. We have revised the corresponding sentence in our manuscript (L59-62).

4. In their analysis, the authors use different 500 different parameter settings to infer the ancestral state for their orthologous groups (OG). They then assign a binary vector of length 3 to each OG, representing the presence of the corresponding gene at three internal nodes according to each of the 500 replicates. They then state that they selected the most strongly supported vector across the replicates to represent the OG. I think what is missing here is an information about the dependencies between the parameter values used for the ancestral state reconstruction. If a subset of the analyses were performed with values drawn from a certain region in the parameter hyperspace, then the outcome of the inferences will be likely similar. Moreover, I interpret Fig. 2c such that the results of the ancestral state reconstruction remain largely unchanged when modifying the parameter values. How does this reconcile with the statement in the main text “Because the estimation of ancestral gene content was uncertain”, which suggests that results change when parameter values are modified?

Thank you for your question. The parameters of the Mk model for the 500 replicates were estimated based on the gene presence/absence data of the extant species and the reference tree by a Bayesian method. We utilized the Markov Chain Monte Carlo (MCMC) method to sample the 500 parameters following their posterior distribution. Therefore, the stable estimation result in **Fig. 2c** suggests that the gene content reconstruction result will be robust against the fluctuation of Mk model parameters. We also showed there is no clear correlation among the parameters of 500 replicates in **Supplemental Figure 1a**, suggesting the sampling interval of MCMC was sufficiently large.

5. Bacteria are renown for sharing genes via recombination and HGT. Such events are not considered here. Moreover, many of the metabolic genes are organized in functional gene clusters. How does this influence the outcome of this analysis?

Thank you for pointing out horizontal gene transfers (HGTs) and the existence of gene clusters. HGTs would have occurred within Lactobacillaceae and affected the gene gain rates of many orthologs. We considered the effects of HGTs by optimizing the gene gain and loss rates simultaneously before ancestral state reconstruction. As you mentioned, genes in the same gene cluster can be lost simultaneously by large deletions of genomic regions as previously suggested (PMID: 21379323; 20523904). Our gene-loss analysis also considers those events in terms that different orthologs can be estimated to be lost at the same branch.

6. Lines 157-159. I do not follow the argumentation. However, this seems important for the study. Please explain and rephrase

We thank you for this comment. **Supplemental Figure 1b** supported the convergence of the metabolic gene repertoire of FLAB, because the two FLAB genera (indicated by green shade) are clustered while the two genera are distantly related in the reference phylogeny. On the other hand, the convergence of non-metabolic gene repertoires was not apparent in the dendrogram, because the two FLAB genera were not clustered (**Supplemental Figure 1c**). We clarified these points in Result section (L184-189).

7. Fig. 3 a represents a standard permutation test to come up with an empirical distribution how similarities of two loss orders are distributed by chance. I do not think that the manuscript benefits much from this figure, and they could as well

be placed into the supplement.

Thank you for your suggestion. As Reviewer 4 requested clarity of explanation about the statistical testing of Figure 3a, we still think the figure can be left in main figures, considering the broad readership of Communications Biology.

8. The search for *adhE* homologs is confusing. Why is it that the authors did not use standard ortholog searches to determine the phylogenetic profile of this gene? What is the rationale to first search for *adhE* like proteins in diverse bacterial phyla returning then to their focal organisms to perform the ortholog search? Moreover, it should be readily accessible if this search was done indeed on the gene level, as suggested by the text, or by analysing the predicted protein sets. When it comes to establishing phylogenetic profiles of proteins across large taxon collections keeping track of changes in ortholog length and feature architecture differences, the authors may want to take a look at [10.3389/fmicb.2021.739000](https://doi.org/10.3389/fmicb.2021.739000) and [10.1371/journal.pgen.1010646](https://doi.org/10.1371/journal.pgen.1010646)

Thank you for asking this. We conducted standard ortholog searches by using a Hidden Markov Model (HMM) but also homologous sequence searches and gene phylogeny estimation. This multi-step approach was adopted because we needed to carefully detect previously reported partial *adhE* genes (PMID: 33563209), which can be missed by standard methods or HMM-based gene search alone. Therefore, we first conducted an HMM-based gene search and then used the hits as queries for the following sequence search by MMSeqs by allowing it to detect short partial sequences. We conducted an HMM search for all bacterial phyla, considering the possibility that *adhE* can be horizontally transferred from non-Lactobacillaceae to Lactobacillaceae species. Lastly, we conducted a phylogenetic analysis to discriminate AdhE-derived ALDH- or ADH-only fragments from other single-domain proteins having ALDH or ADH domains. We elaborated on the explanation of these procedures and the rationale of the analysis design in the Result section (L267-282).

9. The state transition model shown in Fig. 4f is interesting. However, it has to be way better integrated into the story in order to provide a relevant contribution. Moreover, many of the transitions in the graph remain unexplained. Looking closely, it appears as if it would allow transitions that require two independent changes. Is this intentional? If so, then this has to be motivated and explained. On top of this, the model has a transition from “no *adhE*-derived” gene to “complete”. What kind of event is this? A lateral acquisition of a gene? Lastly, why is it that the authors talk about an “*adhE*-derived gene” instead of *adhE*?

Thank you for your interest in that data. **Figure 4f** contributes to the whole story by providing information on quantitative rates of domain losses. As you asked, we intentionally allowed transitions in which both ALDH and ADH domains are lost simultaneously because those transitions can occur by single mutations (e.g., nonsense mutations at upstream regions). Thanks to your comment, we noticed that the rate of a transition from “no AdhE” to “one complete AdhE” was non-zero, suggesting *adhE* genes could be horizontally transferred. We clarified these points in our revised manuscript (L364-371). For clarity, the term “*adhE*-derived genes” represents complete and partial *adhE* genes throughout our manuscript. We used the term because we intended to analyze transitions between complete and partial *adhE* genes in this analysis.

10. I am missing a functional intuition why certain bacterial lineages have a high preference of retaining the ALDH domain rather than the ADH domain. I would be surprised if there is no considerably simple explanation. Naively, I would assume that an orphan ALDH domain can be, with considerably few adaptations be re-used in a different functional context. This might not be possible for the ADH domain.

We highly appreciate this comment. Our hypothesis to explain the order of domain loss is that ALDH fragments are less toxic than ADH fragments. According to a previous study, the acetaldehyde-producing activity of the ALDH domain was decreased by disrupting the polymerization of AdhE, while ethanol-to-acetaldehyde conversion activity of the ADH domain was not affected (PMID: 31586059). Thus, we reasoned that ALDH fragments would have less activity to produce toxic acetaldehyde than ADH fragments after AdhE lost ADH fragments and polymerization activity. We clarified these explanations in the revised manuscript (L468-475).

Minor issues

1. Figure 1. Please correct Sifferent to Different

Thank you for noticing the typo. We corrected it (**Figure 1**).

2. Line 77. The authors may want to revise their statement that FLABs grow poorly on fructose

Thank you for pointing it out. FLAB grow poorly on glucose but well on fructose. We corrected the sentence (L85-86).

3. the more appropriate reference of the absence of AdhE in FLAB might be 10.1007/s00284-013-0506-3

Thank you for your suggestion. We also cited the paper you introduced (L105).

4. Line 119. The authors may want to comment on the selection of a representative genome for each of the 344 species, and how the selection of a different representative may change conclusions.

Thank you for your suggestion. As reviewer 2 also asked about the point, we clarified the criteria of the species selection (L136-140).

5. Figure 2a. The third taxon from the bottom displays an ortholog for “b”. In the ancestral state reconstruction, this is ignored. What is the rationale here? Is this ortholog considered to be a xenolog instead, and based on what evidence?

Thank you for your question. For clarity, this panel represents a schematic example of ancestral state reconstruction, not an actual result. In this example, the ortholog “b” is considered to be a xenolog as you assumed. If there were no horizontal gene transfers, we can assume the latest common ancestor of all the species in the tree possessed the ortholog “b” and the ortholog was repeatedly lost three times (three evolutionary events in total). On the other hand, we only need to assume two gene gains (two evolutionary events) in the scenario presented in Figure 2a, so this scenario is more parsimonious. Notably, we used a maximum likelihood method for ancestral state reconstruction in our actual experiments, and the result was dependent on gene gain and loss rates estimated for each ortholog by a Bayesian method. We elaborated our explanation of the legend of the panel (L159-164).

6. Line 116. How many orthologous groups were analysed? This info is given in Fig 2C but not in the main text.

We analyzed 2,293 orthologs in total as we clarified in our revised manuscript (L141).

7. Line 124. The authors may want to use LCA instead of CA, just for the sake of precision

Thank you for your suggestion. We replaced “CA” with “LCA” throughout the manuscript.

8. Line 130. The wording “robust among” should be revised

Thank you. We corrected the word “among” into “against” (L154).

9. Line 151. Consider rephrasing to “larger than expected by chance”. Either wording, however, requires the result of a

statistical test

We appreciate your suggestion and changed the phrase (L181). We also added information on the statistical test results (L173).

10. Line 154. What is the difference between a “pressure” and a “constraint”. Moreover, either word likely requires the addition of “selective”

Thank you for your suggestion. Selective pressure is an effect that makes individuals showing particular traits more adaptive. On the other hand, evolutionary constraints are physical, genetic, or developmental limitations of traits (PMID: 11234024).

11. Line 156. Please rephrase this sentence. Genes cannot converge in this analysis. It is the composition of gene sets that can converge.

We corrected the wording as you suggested. Thank you. (L185)

12. Line 226. Please introduce GTDB upon its first use

Thank you. We added a brief introduction to GTDB (L134-136).

13. Line 228. Please explain how you used 4,833 *adhE* like genes as query for homology searches. This should be already understandable from reading the main text

This comment is related to your major comment 8. In our study, we first prepared reference *adhE* sequences (4,833 genes) by Hidden Markov Model (HMM)-based sequence searches against all phyla of bacteria by KofamScan and excluding hits shorter than 800 amino acids. Then we set those 4,833 sequences as queries for the following sequence similarity search against Lactobacillaceae genomes by MMseqs to sensibly detect partial genes with thresholds of 45% coverage and 45% aa identity. We elaborated the explanation in the Result section (L272-273)

14. Line 230. Please correct it to “substantially shorter”

Thank you. We corrected it (L282).

15. Line 237. Please revise sentence. As it stands, it is hard to grasp.

Thank you for your suggestion. Reviewer 4 also noted the same point (Comment 14). We split the sentence into two sentences (L283-285).

16. Line 240. The shotgun metagenomics analysis comes out of nowhere and needs a better integration.

We agree with your suggestion. We added appropriate explanations to connect the metagenomic analysis results with the isolation site results (L297-301).

17. Line 248. I think it is not appropriate to say that two taxa “share evolutionary pressure”. Rather they are subjected to the same selective pressure

Thank you very much. We changed the phrase to “were subjected to similar selective pressure” (L305).

18. Line 281. The word “rigorous” generates a wrong impression. Instead, using “sensitive” would be a better option.

We changed the word as you suggested (L319). Thank you for pointing it out.

19. Line 295. If the authors want to make a claim that ALDH-only AdhE proteins are a major evolutionary intermediate,

then Fig4d is not optimal to support this claim. Instead, it would be necessary to show that an ancestral state reconstruction reveals a fragmented AdhE at the internal nodes connecting species that have either lost AdhE or still retain a truncated version.

Thank you for your suggestion. In **Figure 4f**, the substantial transition rates from “Complete × 1” to “ALDH × 1” and from “ALDH × 1” to “No *adhE*” support the claim directly. Furthermore, we thoroughly checked the Figure 4d and found that *Apilactobacillus apinorum* and *Lacticaseibacillus thaliandensis* are completely missing *adhE*, while their sister groups possess one ALDH fragment only. This observation also supports ALDH fragment was the intermediate state of *adhE* loss in their lineages. We added this point in L351-353.

20. Line 311. What is the relevance of the half sentence “or common domain-loss order”?

We deleted the latter half sentence to avoid any misunderstanding (L374). Thank you.

21. Line 325. It is better to replace “fixed” which is a term from population genetics by “invariable” or “conserved”. Moreover, the argumentation here appears circular. 15 sites are selected because they vary across the ALDH-only fragments but are conserved in the complete *adhE* genes. In the next sentence, the authors then state as a result that amino acid composition indeed shows that they are less conserved. I regret that I cannot see the relevance of this statement.

We appreciate your suggestions. We changed the word “conserved” instead of “fixed” (L388). **Figure 5c** and **d** are consistent results, but we can see amino acids in detail only in **Figure 5d**. We revised our manuscript to emphasize this point (L388-389).

I hope that my comments help in improving the manuscript.

Reviewer #4 (Remarks to the Author):

Konno et al. present a comprehensive phylogenetic analysis of a large number of lactic acid bacteria to determine whether convergent evolution follows stochastic or deterministic paths. They showed that distinct lineages of fructophilic lactic acid bacteria have experienced shared losses of orthologs and gene loss. Moreover, they have also identified that a loss of gene *adhE* led to the convergent evolution of FLAB which followed a specific evolutionary path in multiple lineages of lactic acid bacteria sharing fructose rich habitats. This work brings new insights into convergent evolution across multiple biological levels, ranging from amino acid to a protein domain. I believe this work will be of interest to the researchers working in the field of microbial evolution and phylogenetics. Overall, the manuscript is convincing: experiments are well designed and analyzed, the level of detail in the 'Methods' section is appropriate, and the conclusions are supported by the data. While I'm familiar with some of the experiments performed in this manuscript, I don't have much experience handling large-scale phylogeny data. So please forgive my lack of clarity if it comes up in places.

We would like to thank all the constructive comments. We addressed all the suggestions and thoroughly revised the manuscript. In particular, we elaborated our explanation of convergent evolution in the Introduction, and of the statistical test on the intermediate evolutionary processes toward convergence in the Result section. We hope our revision can fully address your comments.

The idea that long-term evolutionary processes share evolutionary paths at multiple levels is quite an interesting topic for investigation. However, I felt that the idea of convergent evolution needs to be explained much more clearly. For example, how and why convergent evolution is deterministic? Can you define the separate evolutionary events leading to multiple lineages? (authors could cite these references: Stayton, T. 2015 and Stern, D. 2013)

Thank you for your suggestions and for introducing papers, which we cited in our revised manuscript. Convergent evolution can deterministically occur because similar selective pressure, which is a deterministic driver of evolution (PMID: 30409860), facilitates propagation of individuals having similar traits in distinct lineages. In the present study, independent losses of the same set of orthologs at different branches in the reference phylogeny were investigated by ancestral state reconstruction of gene content. In two clades of fructophilic lactobacilli (FLAB), we observed independent gene losses of significantly overlapped ortholog sets since their latest common ancestor. Please note that we avoided using the term “deterministic” in our revised manuscript reflecting comments from reviewer 3. We elaborated the explanation in L50-52 and L180-184.

I believe the model presented here needs further explanation, clear supporting evidence from previous studies and how these evolutionary models actually test your hypothesis. Authors have listed a few studies that support the presented evolutionary models. However, the rationale behind adopting these two alternating strategies remains unclear.

We deeply appreciate this comment. In the present study, we aimed to investigate randomness and constraints on intermediate evolutionary steps and processes toward convergence. Those constraints on evolutionary paths have been observed by comparing distinct evolutionary paths toward convergence, i.e., the orders of multiple evolutionary events in different lineages (PMID: 31786211; 31510665; 30171232; 27135164). There are two polarized scenarios when we compare the two distinct evolutionary paths toward convergence: the order of independent but common evolutionary events is different or similar. Thus, we defined two models, the “evolutionary funnel model” and the “evolutionary stream model” to explain those two cases, respectively. The evolutionary funnel model suggests that there are potentially diverse evolutionary

intermediates from the common ancestor to the converged evolutionary outcomes, while the evolutionary stream model suggests that evolutionary constraints allow specific evolutionary paths only. We clarified this point in the Introduction section (L59-L65).

Line 84: The text explaining the result needs further clarification. This seems like an important result, but it was hard to understand the explanation of how the order of gene loss supports the deterministic evolutionary stream model.

Thank you for your comment. To statistically assess whether the order of gene loss was shared between the paths leading to the FLAB clades, we evaluated the similarity between the gene loss orders of the two FLAB lineages. The loss-order similarity was defined as the ratios of ortholog pairs lost in the same order between the two lineages in all pairs of the commonly lost 137 orthologs. We then calculated the null distribution of the loss-order similarity by randomly shuffling the gene-loss orders. Finally, we compared the observed similarity with the null distribution to test whether the observed score is significantly higher than expected by chance (**Fig. 3a**). We explained this point in detail in the Result section (L211-217).

Please find more precise comments below:

1) Line 40: Are authors trying to establish FLAB as a model system? I found it very difficult to extract any meaning from this sentence.

Yes, we tried to present FLAB as model clades to study the intermediate steps of convergent evolution.

2) Line 44: “Both” is misleading in the sentence since the authors talk about three key aspects driving the evolution.

Thank you for your suggestion. We changed the first sentence of the Introduction through our revision. Thanks to comments from you and Reviewer 3, we thoroughly revised the Introduction section to clarify previous studies this work is based on.

3) Line 46: The statement “Convergent evolution is a powerful clue for revealing the deterministic nature of evolution because independent but common evolutionary outcomes strongly suggest the existence of shared selective pressures or constraints” needs citation.

Thank you for your suggestion. We revised the Introduction section and added citations for the corresponding part (L47-50).

4) Line 48: The sentence talks about previous studies, but those studies haven’ been mentioned either as a citation or as an example.

Thank you for your suggestion. We revised the Introduction section and added citations for the corresponding part (L56-57).

5) Fig. 1 Figure panels should have subheadings a) Evolutionary funnel model b) Evolutionary steam model
“Sifferent evolutionary paths” Is this a typo? I guess it has to be “different evolutionary paths”

Thank you very much for notifying. We corrected the typo (**Fig. 1a**).

6) Line 76: The sentence is misleading and contrary to the definition of FLAB (Endo et al 2018). Can they really grow poorly on fructose?

Thank you for pointing it out. FLAB grow poorly on glucose but well on fructose. We corrected the sentence (L85-86).

7) Line 79: I found it very hard to extract any meaning from this sentence. I think there is too much information packed in a single sentence. I recommend you unpack it into a few different sentences so that you can clearly express these ideas.

We appreciate this comment. We unpacked the sentence into two sentences (L85-87).

8) Line 100-107: These are results and should not be included in the introduction.

Thank you for your suggestion. We removed the result contents from the Introduction section (L114-122).

9) Fig. 2a: You need to specify what does a and b orthologs represent.

Thank you for your comment. We described the illustrated example case of the ancestral state reconstruction in detail (L159-164).

10) Fig.2d: does this contingency table need statistical analysis?

We clarified the P-values of the statistical test (L173).

11) Fig.3b: Can you please specify what is the significant p value threshold?

Here we set the P-value threshold as 0.05.

12) Fig.3b: What does “ALL” signify?

The word “all” signifies that Figure 3b is based on the loss order similarity of all the 137 orthologs commonly lost in FLAB lineages. We removed the word in the revised manuscript to avoid any confusion.

13) Fig.3d: It is unclear what the size of the dot indicates here. It would be good to mention this in the figure legend.

The dot size represents the number of orthologs lost at each branch. We added an explanation in the figure legend (L260-261).

14) Line 232: This sentence is hard to understand. I would recommend breaking it into two separate sentences.

Thank you. We separated the part into two sentences (L283-286).

15) Line 388: I believe the use of the word “Instead” is redundant here.

Thank you. We removed the word “instead” (L454).

Reviewers' comments:

Reviewer #2 (Remarks to the Author):

The manuscript has improved but one major aspects and few minor pieces remain to in need of improvement:

- The manuscript should emphasize more strongly that, with possible exception of the loss of AdhE, many of the results described pertain to the transition to an insect-associated lifestyle but not to the transition to FLAB

minor pieces:

- the term "FLAB" should be more rigourously and clearly described.

- With exception of Figure 2b, which shows only a subset of lactobacilli, phylogenetic trees shown in this manuscript confirm the topology of phylogenetic trees of lactobacilli that were published in the past years.

- the role of the two-domain AdhE as acetaldehyde detoxifying enzyme whenever acetyl-CoA is converted to ethanol can be better described.

Specific comments.

Lines 47 to 80. This is a very long introduction on convergent evolution that is not matched by a corresponding discussion section.

line 83. Please define FLAB better – is the definition a phylogenetic / taxonomic definition or based on metabolism? (heterofermentative lactobacilli that require electron acceptors for growth due to the lack of the two-domain AldE)

line 105. As documented in the present study and elsewhere, the lack of AldE is not unique to fructophilic lactobacilli.

Figure 2 and throughout. As indicated before, for both Apilactobacillus and Fructobacillus, the loss of genes (and the reduction of the genome size) that was associated with the transition to an insect / flower associated lifestyle (A3 and F6) is much greater than the loss of genes associated with the loss of AldE – this should be indicated more clearly in results and discussion section.

Figure 2 and throughout. The usefulness of these analyses may depend on the definition of "fructophilic". If the definition is based on phylogeny (fructobacilli and most apilactobacilli except *A. ozonensis*), then the analyses is warranted, if the definition is based on metabolic traits (all heterofermentative lactobacilli that do not ferment pentoses and grow on glucose only in presence of external electron acceptors), it may not be although the two definitions may overlap.

Figure 2 and throughout. Most core genome phylogenetic trees of the Lactobacillaceae published since 2015 and all virtually all trees published since 2020 show that homofermentative organisms and heterofermentative organisms are connected by one node only, this specifically includes the trees shown in Figure 2g and 4a of the present manuscript. The somewhat odd topology of the tree in Figure 2b is thus not based on the number of core genes that were used but based on the selection of genomes as *Pediococcus* was chosen as only homofermentative representative. This limits the usefulness of Figure 2 for any discussion in relation to the transition from homofermentation to heterofermentation but this is obviously not the main point here.

line 172. The genera Fructobacillus and Apilactobacillus also have among the smallest genome sizes in the Lactobacillaceae so the congruent gene loss may relate to that?

line 200. for niche adaptation of lactobacilli, see Duar et al., 2017, lifestyles of lactobacilli: beer and sourdough harbor fructilactobacilli but are not establishment niches that would support evolutionary adaptation.

line 203. Well yes, adaptation to insects / flowers and host adaptation is known to be linked to a reduced genome size.

line 241. As above – two of the three core genome phylogenetic trees match the topology of other trees of the Lactobacillaceae so the selection of genomes for Figure 2a with only one

homofermentative genus but not the selection of genes accounts for the somewhat odd topology. Figure 3d. as above. The biggest loss of genes was associated with the transition from free-living to host adapted lifestyles (A3 and F6) but not with the transition to become FLAB?

line 287. In the lactobacilli, three enzymes produce acetyl phosphate or acetyl-CoA, phosphoketolase, pyruvate formate lyase, pyruvate dehydrogenase, and consequently the need to convert acetyl CoA to ethanol without releasing acetaldehyde.

line 289. The observation that heterofermentative lactobacilli generally lack pyruvate formate lyase was first reported in 2015.

line 300/301. The narrow carbohydrate utilization spectra of insect adapted lactobacilli was noted earlier (doi:10.1093/gbe/evz136).

line 306. If enzymes generating acetaldehyde pose the selective pressure for maintenance of the two domain AdhE, then these enzymes but not "fructophily" are involved. Fructose conversion to mannitol in heterofermentative lactobacilli differs substantially from those metabolic pathways that relate to pyruvate dehydrogenase or pyruvate formate lyase.

line 309. Objected – multiple other insect adapted lactobacilli including the insect adapted *Lactobacillus* species, *Bombilactobacillus*, *Convivina* and *Fructilactobacillus* maintained AdhE

Figure 4g the x-axis label is unclear. If most strains that lost one of two AdhE domains maintained the alcohol dehydrogenase domain which detoxifies acetaldehyde, this makes a case for the link of the two domain enzyme being responsible for detoxification?

line 442. FLAB share their habitat, insects and flowers, with non-FLAB so the selective pressure is not necessarily imposed by the habitat.

line 460. The current manuscript provides two nice examples that the matter of the topology of the phylogenetic tree is settled for good (but that trees can have a deviating topology when a subset of genomes is selected).

line 470. The role in detoxification is supported by several other pieces of evidence (see comments above).

line 490. The loss of AdhE defines FLAB and the term thus also includes homofermentative FLAB? As above, what defines FLAB must be indicated more carefully.

line 500/501. An investigation of the adaptation to fructose-rich environments would have used all insect-associated lactobacilli and not only the FLAB.

Reviewer #3 (Remarks to the Author):

In the revised version of their manuscript, Konno and colleagues have addressed all of my questions. I particularly appreciate the omission of the word 'deterministic', and I think the message stands out more clearly now. I just have a number of minor comments, most of which are just suggestions for different wording. However, it would be great if the authors would have another go on the abstract. I think there is still some room for improving the precision in conveying the main findings of this study. Once again, I hope that my comments help to further improve the manuscript.

Kind regards,
Ingo Ebersberger

Minor comments

1. Abstract Lines 35-36: The authors might consider to rephrase 'phylogenetic comparative methodologies' to something that is more accessible.
2. Line 36 – I think it should read 'we show that the ...'
3. Line 38 – the statement 'likely followed significantly similar orders' needs to be rephrased. Either the observation is statistically significant, then the 'likely' has to be removed, or the order is likely to be the same, then the observation is not statistically significant. However, both together does not

work.

4. Lines 38 and 39 – The word 'suggest' in line 38 already indicates that this is only one of several explanations. I don't think that this statement has to be additionally weakened by 'tended' in line 39. If the signal would be so faint, then it should not be mentioned in the abstract.

5. Line 138 – I suggest to rephrase the statement 'we tried to include the genomes of phylogenetically related clades (...)' for two reasons. First, all bacteria are phylogenetically related to FLAB, so the phrase is too unspecific. Second, and this relates to my next point, if these taxa are really relevant for the study, then one could have tried harder to include them, e.g., by reducing the acceptance threshold taking care to not deteriorate the quality of the data set.

6. Reviewer 2 suggested to complement the taxon set with representatives of close relatives to the FLAB group (*Acetilactobacillus*, *Philodulcिलactobacillus*, and *Nicoliella*). The authors state that no genomes were available meeting their quality criteria (>95% completeness; <5% contamination). These values are, however, ad hoc, and it might be that a slight amendment of the cutoffs would help to have these taxa represented. At least, the authors should provide the values for the two measures for these taxa. It would be great if they could then reason how, for example, a moderately reduced completeness value would bias rather than benefit the analyses.

7. Line 163 – the term 'related' should be omitted. As stated above, all bacteria are related. If the degree of relatedness is relevant, then this should be stated instead.

8. Line 208 – The heading 'gene-loss evolution' should be fixed. It reads strange.

9. Line 254 – Species and genus names should be written in italics. Is there a rationale to have *Pediococcus* in normal font?

10. Line 271 – I trust that it should read profile hidden Markov model (pHMM)

11. Line 359 – 'evolutionary order' should be changed to either 'temporal order' or just 'order'

12. The authors specify that they selected *adhE* genes in the length range between 800 and 1000 amino acids (Methods, line 559).

a. What is the justification of this length range?

b. For the sake of precision, gene length should be given in nt and not in aa. If they want to stick with aa as a unit, then the sentence should be re-written to something like 'we selected genes encoding proteins in the length range between...'

13. To increase precision, the authors should make clear throughout the manuscript when they analyse genes and when they analyse the amino acids sequences encoded by these genes.

14. Line 570 – the authors should indicate what they consider the reference point 'upstream' and 'downstream' relates to

15. Figure titles: The authors may consider giving the individual title that does not describe an analysis process, if the figure shows the outcome of the analysis. The heading of Fig. 2 can serve as an example where an improvement would be great.

Minor discretionary issues

1. Line 37 – I think 'shared losses' should better read with 'parallel losses' to put emphasis on the aspect that the same genes were lost independently on two evolutionary lineages.

2. Line 270 – better, sequence similarity-based search?

Reviewer #4 (Remarks to the Author):

All the major concerns have been addressed in the revised manuscript.

Response to reviewers' comments

For “Evolutionary paths toward multi-level convergence of lactic acid bacteria in fructose-rich environments” (COMMSBIO-23-3101A)

Revision Summary:

We sincerely appreciate the constructive peer-review process to improve our study on the convergent evolution of lactobacilli in fructose-rich environments. We again thoroughly addressed the additional comments raised by the reviewers this time and updated the manuscript. We highlighted all the changes to our manuscript with **yellow markers**. No changes were made to the figures in this revision.

Reviewer #2 (Remarks to the Author):

The manuscript has improved but one major aspects and few minor pieces remain to in need of improvement:

Thank you for all of your comments and questions.

Major comment:

1. The manuscript should emphasize more strongly that, with possible exception of the loss of AdhE, many of the results described pertain to the transition to an insect-associated lifestyle but not to the transition to FLAB

Thank you for your comments. The evolutionary paths observed in this study are not the paths within FLAB clades but the paths toward the latest common ancestor of each FLAB clade (LCAf or LCAa) from the latest common ancestors of the two FLAB clades (LCAfa). Thus, the evolutionary paths include not only the transitions to FLAB but also the transitions to insect-associated lifestyles, as you clarified. We discussed the life-style transitions as a potential cause of the gene losses observed in L207-208 and L458.

Minor comments:

2. the term FLAB should be more rigourously and clearly described.

We agree on the importance of the definition of FLAB in this study. FLAB has been defined as a group of lactobacilli sharing the following metabolic traits: (1) Growing well on fructose and requiring electron acceptors (e.g., fructose, oxygen, or pyruvate) to grow on glucose, (2) Metabolizing limited types of sugars, (3) Lacking many genes for glycometabolism, and (4) Lacking *adhE* fully or partially. We clarified this point in L80-81 and L113-117.

3. With exception of Figure 2b, which shows only a subset of lactobacilli, phylogenetic trees shown in this manuscript confirm the topology of phylogenetic trees of lactobacilli that were published in the past years.

Thank you for your careful confirmation of the figure. The phylogenetic trees except Figure 2b are the whole Lactobacillaceae phylogeny provided by Genome Taxonomy Database (GTDB).

4. the role of the two-domain AdhE as acetaldehyde detoxifying enzyme whenever acetyl-CoA is

converted to ethanol can be better described.

We appreciate this comment. It is certainly possible that the two-domain AdhE works as an acetaldehyde detoxifying enzyme. We clarified this point in L491-494.

Specific comments.

5. Lines 47 to 80. This is a very long introduction on convergent evolution that is not matched by a corresponding discussion section.

Thank you for your comments. As a response to previous feedbacks from Reviewer 3, we have enriched the introduction on convergent evolution. We believe that this revised introduction will be beneficial given the broad readership of Communications Biology unless the editors request that we shorten it.

6. line 83. Please define FLAB better. Is the definition a phylogenetic / taxonomic definition or based on metabolism? (heterofermentative lactobacilli that require electron acceptors for growth due to the lack of the two-domain AldE)

As we respond to Comment 2 of Reviewer 2, we agreed on the importance of the definition of FLAB in this study. FLAB has been defined as a group of lactobacilli sharing the following metabolic traits: (1) Growing well on fructose and requiring electron acceptors (e.g., fructose, oxygen, or pyruvate) to grow on glucose, (2) Metabolizing limited types of sugars, (3) Lacking many genes for glycometabolism, and (4) Lacking *adhE* fully or partially. We clarified the point in L80-81 and L113-117.

7. line 105. As documented in the present study and elsewhere, the lack of AldE is not unique to fructophilic lactobacilli.

Thank you. We removed the word “unique”, because there are non-FLAB species missing *adhE* (L104).

8. Figure 2 and throughout. As indicated before, for both Apilactobacillus and Fructobacillus, the loss of genes (and the reduction of the genome size) that was associated with the transition to an insect / flower associated lifestyle (A3 and F6) is much greater than the loss of genes associated with the loss of AldE. This should be indicated more clearly in results and discussion section.

We appreciate your comments. As we respond to Comment 1 of Reviewer 2, the evolutionary paths observed in this study are not the paths within FLAB clades but the paths toward the latest common ancestors of each FLAB clade (LCAf or LCAa) from the latest common ancestors of the two FLAB clades (LCAfa). Thus, the evolutionary paths include not only the transitions to FLAB but also the transitions to insect-associated lifestyles, as you clarified. We discussed the life-style transitions as a potential cause of the gene losses observed in this study both in Results (L207-208) and Discussion (L458).

9. Figure 2 and throughout. The usefulness of these analyses may depend on the definition of fructophilic. If the definition is based on phylogeny (fructobacilli and most apilactobacilli except *A. ozonensis*), then the analyses is warranted, if the definition is based on metabolic traits (all heterofermentative lactobacilli

that do not ferment pentoses and grow on glucose only in presence of external electron acceptors), it may not be although the two definitions may overlap.

We deeply thank you for this comment. In response to Comments 2 and 6 of Reviewer 2, FLAB has been defined as a group of lactobacilli sharing the following metabolic traits. Based on the definition, previous studies (Maeno et al. 2021) and our analyses (Supplementary Fig. 1b) showed the distinct clades of FLAB have convergent gene repertoire. Therefore, we verified whether gene losses during the genomic convergence occurred in a similar or different order. Although FLAB has been defined as a group of lactobacilli sharing the following metabolic traits, our analysis is valuable in discussing the constraints of long-term genome convergence (Figures 2 and 3)

10. Figure 2 and throughout. Most core genome phylogenetic trees of the Lactobacillaceae published since 2015 and all virtually all trees published since 2020 show that homofermentative organisms and heterofermentative organisms are connected by one node only, this specifically includes the trees shown in Figure 2g and 4a of the present manuscript. The somewhat odd topology of the tree in Figure 2b is thus not based on the number of core genes that were used but based on the selection of genomes as *Pediococcus* was chosen as only homofermentative representative. This limits the usefulness of Figure 2 for any discussion in relation to the transition from homofermentation to heterofermentation but this is obviously not the main point here.

Thank you for pointing out this. To clarify, the topologies of the phylogenetic trees in Fig 2b, 2g, and 4a are the same and the location of *Pediococcus* in all of the trees was different from the previously reported reference phylogeny (PMID: 26253671). Therefore, we verified the robustness of our results by removing or regrafting the *Pediococcus* clade in our previous revision (L242-251).

11. line 172. The genera *Fructobacillus* and *Apilactobacillus* also have among the smallest genome sizes in the Lactobacillaceae so the congruent gene loss may relate to that?

Thank you for your comment. We think that the gene-content convergence of FLAB is not only caused by genome size reduction because gene repertoires of FLAB genomes are reported not to be similar to those of other related clades with small genomes (PMID: 33563209). Therefore, convergent and massive gene losses toward FLAB certainly contributed to their small genome sizes, but only having small genome sizes cannot fully explain the convergence of FLAB. We clarified this point in L102-104.

12. line 200. for niche adaptation of lactobacilli, see Duar et al., 2017, lifestyles of lactobacilli: beer and sourdough harbor fructilactobacilli but are not establishment niches that would support evolutionary adaptation.

Thank you for your comment and for introducing a key paper. As you pointed out, *Fructilactobacillus* species do not share their habitats while *Apilactobacillus* species inhabit common habitats. Each of *Fructilactobacillus* species, however, shows adaptation to its habitat, such as *F. florum* adapting to fructose-rich environments, *F. sanfranciscensis* adapting to sourdough (e.g., maltose preference), and *F. lindneri* adapting to beer (e.g., hop resistance). Overall, *Fructilactobacillus* species are specialists in their habitat

environments, which is consistent with the fact that they have small genomes because specialists are known to possess smaller genomes than generalists in general (PMID: 29079803).

13. line 203. Well yes, adaptation to insects / flowers and host adaptation is known to be linked to a reduced genome size.

Thank you for letting us know this point. We clarified that a transition to insect/flower environments could drive massive gene losses (L207-208).

14. line 241. As above, two of the three core genome phylogenetic trees match the topology of other trees of the Lactobacillaceae so the selection of genomes for Figure 2a with only one homofermentative genus but not the selection of genes accounts for the somewhat odd topology.

Thank you for pointing out this. As clarified in the response to Comment 10 of Reviewer 2, the topologies of the phylogenetic trees in Fig 2b, 2g, and 4a are the same and the location of *Pediococcus* in all of the trees was different from the previously reported reference phylogeny (PMID: 26253671). When we closely look at Figure 4a, we notice that homofermentative and heterofermentative species are not partitioned by a single branch. We think the topology difference with the previous study is because of the difference in the selection of marker gene sets. We verified the robustness of our results by removing or regrafting the *Pediococcus* clade in our previous revision (L242-251).

15. Figure 3d. as above. The biggest loss of genes was associated with the transition from free-living to host adapted lifestyles (A3 and F6) but not with the transition to become FLAB?

We agree that evolution at the branches A3 and F6 is not with the transition to become FLAB. As we clarified in the Comment 1 of Reviewer 2, the evolutionary paths observed in this study are not only the transition to FLAB clades but all the paths toward the latest common ancestor of each FLAB clade from the latest common ancestors of the two FLAB clades. This is because our objective here was to verify the similarity of the whole long-term evolutionary paths toward convergence. Thus, the evolutionary paths would include not only the transitions to FLAB but also the transitions to insect-associated lifestyles, as you clarified. We discussed the life-style transitions as a potential cause of the gene losses observed in this study in L207-208 and L458.

16. line 287. In the lactobacilli, three enzymes produce acetyl phosphate or acetyl-CoA, phosphoketolase, pyruvate formate lyase, pyruvate dehydrogenase, and consequently the need to convert acetyl CoA to ethanol without releasing acetaldehyde.

Thank you for letting us know this. Although we analyzed the relationship between *AdhE* and pyruvate formate lyase in our study, the phylogenetic distribution of phosphoketolase and pyruvate dehydrogenase can also be interesting to be analyzed in the future. Species missing *adhE* may tend to miss those enzyme genes.

17. line 289. The observation that heterofermentative lactobacilli generally lack pyruvate formate lyase was first reported in 2015.

Thank you for letting us know that report. We cited the paper in our revised manuscript (L295-296).

18. line 300/301. The narrow carbohydrate utilization spectra of insect adapted lactobacilli was noted earlier (doi:10.1093/gbe/evz136).

Thank you for pointing it out. We cited the paper (L204-205).

19. line 306. If enzymes generating acetaldehyde pose the selective pressure for maintenance of the two domain AdhE, then these enzymes but not fructophily are involved. Fructose conversion to mannitol in heterofermentative lactobacilli differs substantially from those metabolic pathways that relate to pyruvate dehydrogenase or pyruvate formate lyase.

Thank you for clarifying the point. As you suggested, the selective pressure on the *adhE* possession in homofermentative species having pyruvate dehydrogenase or pyruvate formate lyase can be different from that in heterofermentative species. There are many homofermentative species, however, missing pyruvate formate lyase but possessing *adhE*, which raises a possibility that there is an unknown selective pressure in homofermentative species. The unknown pressure may be relaxed under fructose-rich environments and be similar to the pressure on heterofermentative species, because *adhE* was shown to be lost in fructose-rich environments both in homofermentative and heterofermentative species.

20. line 309. Objected multiple other insect adapted lactobacilli including the insect adapted *Lactobacillus* species, *Bombilactobacillus*, *Convivina* and *Fructilactobacillus* maintained AdhE

We appreciate your point. Evolution is always stochastic to some extent, so species inhabiting fructose-rich environments do not always lose *adhE*. There might be unexpected selective pressures to maintain *adhE* in some insect-associated species. Our results, however, showed that species detected in fructose-rich environments tend to miss *adhE* more often than expected by chance (L297-301, L303-308).

21. Figure 4g the x-axis label is unclear. If most strains that lost one of two AdhE domains maintained the alcohol dehydrogenase domain which detoxifies acetaldehyde, this makes a case for the link of the two domain enzyme being responsible for detoxification?

Thank you for your comments. We explained the x-axis label corresponds to Fig 4f in the figure legend (L344-345). We agree that ADH-only fragments can be responsible for the detoxification of acetaldehyde if lactobacilli often possess ADH-only fragments derived from AdhE. On the other hand, our results showed that ALDH-only fragments more frequently remained than ADH-only fragments (Fig. 4g).

22. line 442. FLAB share their habitat, insects and flowers, with non-FLAB so the selective pressure is not necessarily imposed by the habitat.

Thank you for pointing it out. Although insect/flower is not a specific habitat to FLAB as you mentioned, insect/flower-associated lifestyle could impose a selective pressure leading to the gene loss observed in the branches A3 and F6. In this study, we intentionally included such ancestral evolutionary processes in our analysis to compare the entire evolution from the common ancestor of the two FLAB (LCAfa) to the common ancestors of distinct FLAB clades (LCAf and LCAa) as we explained responding to Comments 1 and 8 of Reviewer 2.

23. line 460. The current manuscript provides two nice examples that the matter of the topology of the phylogenetic tree is settled for good (but that trees can have a deviating topology when a subset of genomes is selected).

Thank you for your comments. We agree on the importance of the reference species tree topology (L238-247). As we respond to Comments 10 and 14 of Reviewer 2, the topologies of the phylogenetic trees in Fig. 2b, 2g, and 4a are the same, although Fig. 2b shows a part of the whole phylogeny shown in Fig. 2g and 4a.

24. line 470. The role in detoxification is supported by several other pieces of evidence (see comments above).

Thank you for letting us know that. As a response to Comment 4 of Reviewer 2, we discussed the possible roles of complete and fragmental AdhE in detoxification of aldehyde (L491-494).

25. the role of the two-domain AdhE as acetaldehyde detoxifying enzyme whenever acetyl-CoA is converted to ethanol can be better described.

Thank you. As a response to Comment 4 of Reviewer 2, we added a discussion that AdhE might have detoxification activity before losing the ADH domain (L491-492).

26. line 490. The loss of AdhE defines FLAB and the term thus also includes homofermentative FLAB? As above, what defines FLAB must be indicated more carefully.

We deeply appreciate this comment and other related comments (Comments 2, 6, and 9 of Reviewer 2). As we clarified in L80-81 and L113-117, FLAB has been defined as a group of lactobacilli sharing the following metabolic traits, and we adhered to this definition throughout this study. Based on the definition, homofermentative species missing AdhE are not included in FLAB, because homofermentative species do not need electron acceptors (e.g., fructose and oxygen) for glucose metabolism. As homofermentative species metabolize glucose using the Embden-Meyerhof pathway, not phosphoketolase pathway, homofermentative species do not require fructose to recover NAD^+ from NADH.

27. line 500/501. An investigation of the adaptation to fructose-rich environments would have used all insect-associated lactobacilli and not only the FLAB.

Thank you so much for the comment useful for our future study. We used as many genomes of lactobacilli

as possible in this study, but the available high-quality reference genomes (>95% completeness; <5% contamination) were limited for insect-associated lactobacilli. Generating high-quality genome sequence data for insect-associated species and analyzing them by the analysis framework similar to this study is one of our future directions.

Reviewer #3 (Remarks to the Author):

Minor comments

1. Abstract Lines 35-36: The authors might consider to rephrase “phylogenetic comparative methodologies” to something that is more accessible.

Thank you. We changed the words into “Phylogenetic Comparative Methods” (L36).

2. Line 36 I think it should read “we show that the”;

Thank you. We changed the wording as you suggested (L36).

3. Line 38 the statement “likely followed significantly similar orders” needs to be rephrased. Either the observation is statistically significant, then the “likely” has to be removed, or the order is likely to be the same, then the observation is not statistically significant. However, both together does not work.

Thank you. We omitted “likely” (L37).

4. Lines 38 and 39 The word “suggest” in line 38 already indicates that this is only one of several explanations. I don’t think that this statement has to be additionally weakened by “tended” in line 39. If the signal would be so faint, then it should not be mentioned in the abstract.

Thank you. We omitted “tended to” (L39).

5. Line 138 I suggest to rephrase the statement “we tried to include the genomes of phylogenetically related clades” for two reasons. First, all bacteria are phylogenetically related to FLAB, so the phrase is too unspecific. Second, and this relates to my next point, if these taxa are really relevant for the study, then one could have tried harder to include them, e.g., by reducing the acceptance threshold taking care to not deteriorate the quality of the data set.

Thank you for the comment. We added the statement as a response to Reviewer 2 in the previous revision. As you and Reviewer 2 suggested, we agreed on the importance of an explanation of how we selected species to analyze. Therefore, we explained the rationale of setting the genome quality threshold in our revised manuscript as we respond to Comment 6 of Reviewer 3 (L142-144).

6. Reviewer 2 suggested to complement the taxon set with representatives of close relatives to the FLAB group (*Acetilactobacillus*, *Philodulcilitobacillus*, and *Nicoliella*). The authors state that no genomes were

available meeting their quality criteria (>95% completeness; <5% contamination). These values are, however, ad hoc, and it might be that a slight amendment of the cutoffs would help to have these taxa represented. At least, the authors should provide the values for the two measures for these taxa. It would be great if they could then reason how, for example, a moderately reduced completeness value would bias rather than benefit the analyses.

We deeply appreciate this comment and agree that the criteria look ad hoc. We set the thresholds of completeness and contamination to avoid the risk of false prediction in the ortholog presence/absence table we constructed. Intuitively, X% of genome completeness suggests around (100 – X)% of truly possessed orthologs in a genome are mistakenly regarded as absent (false negative). Y% of contamination suggests around Y% of seemingly present orthologs in a genome are derived from other genomes (false positive). We clarified the point in L142-144.

7. Line 163 the term “related” should be omitted. As stated above, all bacteria are related. If the degree of relatedness is relevant, then this should be stated instead.

Thank you. We strictly described the figure as the phylogenetic relationship of “descendants of the latest common ancestor of two FLAB clades” (L168-169).

8. Line 208 The heading “gene-loss evolution” should be fixed. It reads strange.

Thank you. We changed the wording to “Gene loss events toward FLAB clades supported...” (L215).

9. Line 254 Species and genus names should be written in italics. Is there a rationale to have *Pediococcus* in normal font?

We corrected the normal font of all the appearances of “*Pediococcus*”.

10. Line 271 I trust that it should read profile hidden Markov model (pHMM)

Thank you, we corrected the wording (L278).

11. Line 359 “evolutionary order” should be changed to either “temporal order” or just “order”

We changed the word to just “order”, and made the sentence clearer (L369).

12. The authors specify that they selected *adhE* genes in the length range between 800 and 1000 amino acids (Methods, line 559).

12A. What is the justification of this length range?

We decided the length range by looking at the distribution of proteins detected as AdhE by KofamScan (Supplemental Fig. 6d). The range 800-1,000 covers the largest peak of the distribution, and 95.8% of all the detected genes.

12B. For the sake of precision, gene length should be given in nt and not in aa. If they want to stick with aa as a unit, then the sentence should be re-written to something like “we selected genes encoding proteins in the length range between”

Thank you. We used the phrase “genes encoding proteins in the length range between” (L572).

13. To increase precision, the authors should make clear throughout the manuscript when they analyse genes and when they analyse the amino acids sequences encoded by these genes.

Thank you for your question. Throughout this study, we only analyzed amino acid sequences, because GTDB provides the results of gene annotation by Prodigal for every representative genome. We made this point clear by revising the explanation of the dataset (L516).

14. Line 570 the authors should indicate what they consider the reference point “upstream” and “downstream” relates to

Thank you. We used the terms “N-terminal” and “C-terminal” instead (L584).

15. Figure titles: The authors may consider giving the individual title that does not describe an analysis process, if the figure shows the outcome of the analysis. The heading of Fig. 2 can serve as an example where an improvement would be great.

Thank you. We changed some main figure titles to describe the results presented in the figure (Figs. 2-5).

Minor discretionary issues

1. Line 37 I think “shared losses” should better read with “parallel losses” to put emphasis on the aspect that the same genes were lost independently on two evolutionary lineages.

Thank you. We changed the word from “shared” to “parallel” (L37).

2. Line 270 better, sequence similarity-based search?

Thank you. We changed the words to “sequence similarity-based” search (L277).

REVIEWERS' COMMENTS:

Reviewer #2 (Remarks to the Author):

The comments to the last version were adequately addressed.

Reviewer #3 (Remarks to the Author):

I have no more comments.